# The Neuroprotective Role of Curcumin: From Molecular Pathways to Clinical Translation—A Narrative Review

**DOI:** 10.3390/nu17172884

**Published:** 2025-09-06

**Authors:** Andrea Lehoczki, Mónika Fekete, Tamás Jarecsny, Virág Zábó, Ágnes Szappanos, Tamás Csípő, Ágnes Lipécz, Dávid Major, Vince Fazekas-Pongor, Péter Varga, János Tamás Varga

**Affiliations:** 1Institute of Preventive Medicine and Public Health, Semmelweis University, 1085 Budapest, Hungary; ceglediandi@freemail.hu (A.L.); fekete.monika@semmelweis.hu (M.F.); csipo.tamas@semmelweis.hu (T.C.); lipecz.agnes@semmelweis.hu (Á.L.); major.david@semmelweis.hu (D.M.); pongor.vince@semmelweis.hu (V.F.-P.); varga.peter@semmelweis.hu (P.V.); 2Fodor Center for Prevention and Healthy Aging, Semmelweis University, 1085 Budapest, Hungary; zabo.virag@semmelweis.hu; 3Health Sciences Division, Doctoral College, Semmelweis University, 1085 Budapest, Hungary; 4Department of Neurology and Stroke, Saint John’s Central Hospital of North Buda, 1125 Budapest, Hungary; jarecsny.tamas@janoskorhaz.hu; 5Heart and Vascular Center, Semmelweis University, 1122 Budapest, Hungary; drszappanos@gmail.com; 6Department of Rheumatology and Clinical Immunology, Semmelweis University, 1085 Budapest, Hungary; 7Department of Pulmonology, Semmelweis University, 1083 Budapest, Hungary

**Keywords:** curcumin, neurodegeneration, Alzheimer’s disease, Parkinson’s disease, polyphenols, neuroinflammation, oxidative stress, narrative review, clinical translation

## Abstract

Neurodegenerative disorders, including Alzheimer’s disease (AD), Parkinson’s disease (PD), and post-stroke cognitive impairment (PSCI), represent an escalating global health and economic challenge. In the quest for disease-modifying interventions, natural polyphenols—most notably curcumin, the principal bioactive compound of Curcuma longa—have attracted considerable interest due to their pleiotropic neuroprotective effects. This narrative review critically synthesizes findings from a selection of peer-reviewed articles published between 2000 and 2025, chosen for their relevance to curcumin’s molecular mechanisms and translational potential. Curcumin’s complex chemical structure confers antioxidant, anti-inflammatory, and epigenetic modulatory properties; however, its clinical application is limited by poor oral bioavailability. Mechanistically, curcumin attenuates oxidative stress and suppresses key inflammatory mediators, including nuclear factor kappa B (NF-κB), cyclooxygenase-2 (COX-2), and inducible nitric oxide synthase (iNOS). Additionally, it modulates apoptosis, inhibits amyloid-beta aggregation, and enhances cellular quality control processes such as autophagy and mitophagy, while upregulating neurotrophic factors such as brain-derived neurotrophic factor (BDNF). Preclinical studies employing rodent models of AD, PD, and ischemic stroke have demonstrated curcumin’s dose-dependent neuroprotective efficacy, with improved outcomes observed using nanoparticle-based delivery systems. Early-phase clinical trials further support curcumin’s favorable safety profile and potential cognitive benefits, although challenges remain regarding pharmacokinetics, formulation standardization, and therapeutic reproducibility. Future directions include the development of advanced drug delivery platforms, combinatory therapeutic regimens, and personalized medicine approaches integrating curcumin within multifaceted neurotherapeutic strategies. Collectively, this narrative review highlights curcumin as a promising multi-targeted candidate for combating neurodegenerative diseases, while emphasizing the need for further translational and clinical validation.

## 1. Introduction

Neurodegenerative diseases such as Alzheimer’s disease (AD), Parkinson’s disease (PD), and post-stroke neurodegeneration pose a significant and growing global health challenge, with profound social and economic consequences [1]. These disorders are characterized by progressive neuronal loss and functional decline, and current therapeutic options largely focus on symptom management rather than disease modification [2]. Central to neurodegeneration are interrelated molecular mechanisms, including oxidative stress, chronic neuroinflammation, mitochondrial dysfunction, impaired proteostasis, and apoptosis, all contributing to irreversible neuronal damage [3].

Recently, naturally derived polyphenols with pleiotropic bioactivity have garnered considerable attention for their capacity to modulate these interconnected pathological pathways [4,5]. Among them, curcumin—the principal bioactive compound in Curcuma longa (turmeric)—has emerged as a promising neuroprotective agent due to its antioxidant, anti-inflammatory, anti-apoptotic, and metal-chelating properties [6]. Long utilized in traditional Asian medicine, curcumin has recently attracted biomedical interest for its ability to target key molecular drivers of neurodegeneration [7].

Curcumin exerts diverse neuroprotective effects, including modulation of intracellular signaling cascades, preservation of mitochondrial integrity, suppression of microglial activation, and enhancement of synaptic plasticity [8,9]. Notably, curcumin interacts with nicotinic acetylcholine receptors, particularly the α7 subtype (α7-nAChRs), which regulate dopaminergic neurotransmission, neuroinflammation, and cellular resilience [10]. Experimental models of PD suggest that α7-nAChR modulation by curcumin may contribute to improved motor function and dopaminergic neuron survival [11,12].

Despite promising in vitro, in vivo, and in silico evidence, clinical translation is hindered by curcumin’s low oral bioavailability, rapid metabolism, and limited central nervous system penetration [13]. Nevertheless, preclinical and early clinical studies report beneficial effects on cognitive function, neuroinflammation, and neuronal survival across multiple neurodegenerative models [14,15].

This article presents a narrative review that synthesizes and critically evaluates current evidence on the neuroprotective properties of curcumin, with particular emphasis on its molecular mechanisms in neurodegenerative disorders such as Alzheimer’s disease, Parkinson’s disease, and post-stroke cognitive impairment. The literature was identified through a broad search of peer-reviewed publications in major biomedical databases (2000–2025), with selection based on thematic relevance and scientific quality rather than strict systematic criteria. By clarifying the mechanistic basis of curcumin’s neuroprotective effects and assessing its translational potential, this review aims to provide a balanced overview of both the opportunities and limitations of curcumin-based interventions in neurodegenerative diseases.

## 2. Methods

### 2.1. Literature Search Strategy

A comprehensive literature search was conducted between March and May 2025 across four major databases—PubMed/MEDLINE, Web of Science, Scopus, and Embase—supplemented by Google Scholar to capture grey literature and additional sources not indexed in traditional databases.

The search strategy combined Medical Subject Headings (MeSH) with free-text keywords to ensure broad coverage. The primary search strings included:“curcumin” AND (“neurodegeneration” OR “neurodegenerative disease” OR “Parkinson’s disease” OR “Alzheimer’s disease” OR “stroke” OR “cognitive impairment”)“curcumin” AND (“oxidative stress” OR “neuroinflammation” OR “BDNF” OR “NF-κB” OR “mitophagy” OR “apoptosis”)“curcumin” AND (“bioavailability” OR “nanoparticles” OR “clinical trial”)

Searches were limited to English-language publications between January 2000 and May 2025. The temporal restriction was chosen because research on the molecular mechanisms of curcumin in neurodegenerative disorders has significantly expanded since 2000, coinciding with advances in molecular biology techniques, biomarker discovery, and the initiation of early-phase clinical trials. Earlier literature prior to 2000 primarily addressed curcumin’s general pharmacological and anti-inflammatory effects, without a strong focus on neurodegeneration. Thus, this timeframe was selected to ensure both scientific relevance and methodological rigor. No restrictions were placed on study design, enabling the inclusion of in vitro, in vivo, clinical studies, and systematic or narrative reviews. Reference lists of key articles were also manually screened to identify additional eligible studies.

### 2.2. Study Selection and Inclusion Criteria

Studies were included if they:(i)investigated molecular or cellular mechanisms of curcumin relevant to neurodegenerative disorders;(ii)reported preclinical data from in vitro or in vivo models, or clinical evidence of neuroprotective effects;(iii)addressed pharmacokinetics, bioavailability, or delivery strategies within a neurological context.

Exclusion criteria were:(i)non-peer-reviewed publications (unless of significant foundational value);(ii)studies focused solely on non-neurological conditions;(iii)duplicate or non-English publications.

### 2.3. Guiding Research Question

To provide focus and clarity, the following guiding research question was formulated: “What are the key molecular mechanisms by which curcumin exerts neuroprotective effects, and how can these mechanisms be translated into clinical applications in neurodegenerative diseases?”

### 2.4. Data Extraction and Synthesis

For each included study, the following data were extracted:(i)molecular pathways modulated by curcumin (oxidative stress, neuroinflammation, autophagy, apoptosis, mitochondrial function);(ii)pharmacokinetic characteristics and bioavailability enhancement strategies;(iii)experimental models used, particularly animal models of Alzheimer’s disease, Parkinson’s disease, and ischemic stroke;(iv)clinical efficacy and safety outcomes.

The evidence was synthesized narratively, integrating mechanistic insights with preclinical and clinical findings. Special attention was given to identifying translational gaps, methodological limitations, and promising future directions for curcumin-based therapeutic interventions in neurodegenerative disorders.

## 3. Biological Properties of Curcumin

### 3.1. Chemical Structure and Physical Characteristics

Curcumin (diferuloylmethane), the principal polyphenolic constituent of Curcuma longa, was first isolated in the early 19th century, and its chemical structure was elucidated in 1910 [16]. It is a symmetric molecule composed of two aromatic rings substituted with ortho-methoxy and phenolic groups, linked by a seven-carbon chain containing a conjugated β-diketone moiety [17,18]. The molecule exists in tautomeric keto and enol forms, with the keto form predominating under acidic or neutral pH conditions, whereas the enol form is favored in alkaline environments [19].

The curcuminoid fraction of turmeric primarily comprises three naturally occurring analogs: curcumin (I), demethoxycurcumin (II), and bisdemethoxycurcumin (III), each contributing differently to its biological activity [20]. Curcumin is a highly lipophilic compound that is practically insoluble in water but soluble in organic solvents such as ethanol, dimethyl sulfoxide (DMSO), and acetone [21]. Its hydrophobic nature and chemical instability under physiological conditions pose significant challenges for its pharmacological application [22].

### 3.2. Pharmacokinetics and Bioavailability

Following oral administration, curcumin demonstrates poor systemic bioavailability due to limited intestinal absorption, rapid hepatic metabolism, and swift systemic elimination [23]. Clinical pharmacokinetic studies have shown that even high oral doses (4–8 g/day) yield low plasma concentrations, typically ranging from 0.4 to 1.7 micromoles per liter (μmol/L) [24]. These low levels are largely attributed to extensive first-pass hepatic metabolism, including reductive metabolism and subsequent conjugation with glucuronic acid and sulfate, producing curcumin glucuronides and sulfates (Figure 1) [25].

To overcome these limitations, a wide range of formulation strategies has been developed to enhance curcumin’s oral bioavailability [26]. These include nanoemulsions, liposomal encapsulation, and polymer-based nanoparticle systems, such as poly(lactic-co-glycolic acid) (PLGA), alginate, and lactoferrin nanoparticles [27]. Cyclodextrin complexes have also been shown to improve curcumin’s aqueous solubility and chemical stability. In addition, co-administration with piperine (an alkaloid derived from Piper nigrum) significantly inhibits curcumin glucuronidation, thereby increasing its plasma concentration by up to 2000% and extending its elimination half-life [28]. These advanced delivery systems not only increase systemic exposure but also enhance blood–brain barrier (BBB) penetration, which is crucial for achieving therapeutic concentrations in the central nervous system, particularly in the context of neurodegenerative diseases such as Alzheimer’s and Parkinson’s disease [29,30].

### 3.3. Antioxidant, Anti-Inflammatory, and Epigenetic Effects

Curcumin exhibits a broad spectrum of biological activities, with its antioxidant and anti-inflammatory properties underpinning its neuroprotective potential [31]. As an antioxidant, curcumin directly scavenges reactive oxygen and nitrogen species (ROS/RNS) and upregulates endogenous antioxidant enzymes, including superoxide dismutase, catalase, and glutathione peroxidase [32]. It further inhibits lipid peroxidation and preserves mitochondrial integrity under oxidative stress conditions [33].

The anti-inflammatory effects of curcumin are mediated via downregulation of pivotal inflammatory signaling pathways, notably nuclear factor-kappa B (NF-κB), alongside suppression of pro-inflammatory enzymes such as cyclooxygenase-2 (COX-2) and inducible nitric oxide synthase (iNOS) [34]. Additionally, curcumin attenuates microglial activation, thereby mitigating neuroinflammation in pathological contexts [8].

At the epigenetic level [35], curcumin modulates DNA methylation, histone acetylation, and microRNA expression, regulating gene networks associated with neuronal survival, apoptosis inhibition, and synaptic plasticity [36]. These epigenetic modifications contribute to the restoration of cellular homeostasis and enhanced resilience in neurodegenerative milieus [37]. Collectively, curcumin’s capacity to attenuate oxidative damage, suppress neuroinflammation, and promote neuronal survival underscores its therapeutic promise in central and peripheral neurodegenerative disorders [38,39]. Its multi-targeted mechanisms of action render curcumin a compelling natural candidate in the development of effective neuroprotective interventions [11].

## 4. Molecular Mechanisms Underlying the Neuroprotective Effects of Curcumin

The neuroprotective potential of curcumin arises from its capacity to modulate multiple interconnected pathophysiological processes within the central nervous system (CNS) [14,40]. Curcumin is widely recognized for its potent, multifaceted anti-aging [41,42,43], pro-survival, and pro-longevity properties [41,42] exerted via synergistic modulation of diverse cellular and molecular mechanisms associated with aging and neurodegeneration [44]. Its key molecular targets encompass oxidative stress, inflammatory signaling pathways, apoptosis, amyloid protein aggregation, autophagy/mitophagy, and neurotrophic factor regulation.

Emerging evidence suggests that the gut–brain axis plays a critical role in the neuroprotective effects of curcumin, a pathway that has received limited attention in both preclinical and clinical analyses. Given curcumin’s minimal oral bioavailability, a substantial portion of the ingested compound remains in the gastrointestinal tract, where it can directly modulate gut microbiota composition and function. This alternative mechanism provides a plausible explanation for curcumin’s neuroprotective effects independent of systemic plasma levels.

Preclinical studies have demonstrated that curcumin favorably alters gut microbial communities, promoting the proliferation of beneficial taxa such as Lactobacillus and Bifidobacterium, while inhibiting pro-inflammatory bacteria [45,46,47,48]. Curcumin also enhances the production of microbial metabolites, particularly short-chain fatty acids (SCFAs) such as butyrate, which exert systemic anti-inflammatory and neuroprotective effects by modulating microglial activation and tight junction protein expression in the gut epithelium [46,49]. Furthermore, curcumin has been shown to improve intestinal barrier integrity, reducing the so-called “leaky gut” phenomenon that is increasingly implicated in systemic inflammation and neuroinflammation [50,51].

Mechanistically, SCFAs produced under curcumin-induced microbial modulation can influence central nervous system function through several pathways, including histone deacetylase inhibition, enhancement of regulatory T cell differentiation, and modulation of inflammatory signaling via NF-κB and inflammasome pathways. This gut-mediated signaling may complement curcumin’s direct CNS actions, providing a comprehensive explanation for its observed neuroprotective efficacy despite low systemic bioavailability. Integrating these findings offers a more complete mechanistic framework, suggesting that curcumin’s neuroprotective potential is mediated not only by direct antioxidant and anti-inflammatory effects in the CNS but also through modulation of peripheral microbiota–gut–brain interactions [49,52].

### 4.1. Modulation of Oxidative Stress by Curcumin: Molecular Mechanisms and Neuroprotective Potential

Oxidative stress, characterized by the excessive intracellular accumulation of reactive oxygen species (ROS) and reactive nitrogen species (RNS), plays a central role in the etiology of numerous neurodegenerative disorders, including Alzheimer’s disease, Parkinson’s disease, and amyotrophic lateral sclerosis (ALS) [53,54]. The resulting oxidative damage contributes to aberrant protein aggregation, mitochondrial dysfunction, DNA strand breaks, and the activation of neuroinflammatory and apoptotic pathways [55]. Curcumin, a natural polyphenolic antioxidant, mitigates these deleterious effects through multiple mechanisms, including direct radical scavenging, upregulation of endogenous antioxidant defenses, and preservation of mitochondrial function.

#### 4.1.1. Direct Free Radical-Scavenging Activity

Curcumin’s chemical structure, featuring phenolic hydroxyl groups, enables it to function as an effective hydrogen or electron donor, thereby directly neutralizing reactive free radicals such as superoxide anion (O_2_^−^), hydroxyl radical (•OH), peroxynitrite (ONOO^−^), and hydrogen peroxide (H_2_O_2_) [56]. This potent radical-scavenging capacity has been extensively validated in vitro, in animal models, and in clinical contexts [57,58]. For example, in benzo[a]pyrene-exposed murine models, oral administration of curcumin at 60 mg/kg significantly attenuated ROS production and lipid peroxidation in pulmonary tissue, concomitant with restoration of intracellular glutathione (GSH) levels [59]. Similarly, a human intervention study involving subjects with chronic exposure to arsenic-contaminated water demonstrated that supplementation with 500 mg curcumin twice daily markedly reduced systemic ROS, lipid peroxidation, and DNA damage in peripheral blood mononuclear cells [60].

#### 4.1.2. Activation of Endogenous Antioxidant Enzymes

In addition to its direct radical-scavenging activity, curcumin enhances endogenous cellular antioxidant defenses by upregulating the activity and expression of critical antioxidant enzymes. Preclinical studies in rodent models have demonstrated that curcumin significantly increases the activity of superoxide dismutase (SOD), glutathione peroxidase (GPx), glutathione reductase (GR), catalase (CAT), and glutathione-S-transferase (GST) across multiple tissues, including liver, kidney, and nervous system [61,62]. These enzymes play vital roles in maintaining intracellular redox balance by detoxifying reactive oxygen species and preventing lipid peroxidation, thereby contributing to cellular protection against oxidative injury.

#### 4.1.3. Activation of the Nrf2–ARE Pathway

Curcumin also exerts its antioxidant effects through activation of the nuclear factor erythroid 2–2-related factor 2 (Nrf2) signaling pathway, a master regulator of cellular defense against oxidative stress [63,64]. Upon activation, Nrf2 dissociates from its cytoplasmic inhibitor Keap1, translocates into the nucleus, and binds to antioxidant response elements (ARE) in the promoter regions of target genes. This induces the transcription of cytoprotective enzymes such as heme oxygenase-1 (HO-1), NAD(P)H quinone oxidoreductase-1 (NQO1), and glutathione synthetase, which collectively enhance cellular resilience to oxidative damage. Curcumin has been shown to promote Nrf2 nuclear translocation and increase HO-1 expression in neuronal, endothelial, and renal cell models [65,66]. This pathway is especially relevant in the central nervous system, where basal antioxidant capacity is often compromised in neurodegenerative conditions.

#### 4.1.4. Mitochondrial Protection and Inhibition of Lipid Peroxidation

Mitochondria serve both as a primary source and a principal target of oxidative stress. Curcumin preserves mitochondrial membrane integrity, maintains mitochondrial membrane potential, and attenuates mitochondrial ROS generation [67]. Additionally, it stabilizes the electron transport chain complexes, particularly Complexes I and III, and inhibits cardiolipin oxidation, an early event triggering apoptotic cascades [68]. The anti–lipid peroxidation properties of curcumin have been corroborated in various in vivo models, as evidenced by significant reductions in malondialdehyde (MDA) and 4-hydroxynonenal (4-HNE) levels, both established biomarkers of oxidative membrane damage [59,69].

### 4.2. Anti-Inflammatory Mechanisms of Curcumin: Inhibition of NF-κB, COX-2, and iNOS

Curcumin exerts broad-spectrum anti-inflammatory effects predominantly via modulation of the nuclear factor kappa B (NF-κB) signaling pathway [70]. By inhibiting NF-κB activation, curcumin suppresses the transcription of pivotal pro-inflammatory mediators, including cytokines such as interleukin-1β (IL-1β), tumor necrosis factor-alpha (TNF-α), interleukin-6 (IL-6), COX-2, and iNOS. This downregulation reduces the biosynthesis of nitric oxide (NO) and prostaglandins, culminating in attenuation of both systemic and central nervous system inflammation. Consequently, curcumin is regarded as a promising adjunct therapeutic agent in inflammatory and neurodegenerative disorders (Figure 2).

Chronic NF-κB activation is a pathological hallmark of neurodegenerative diseases, driven mainly by sustained microglial activation, oxidative stress, apoptosis, and persistent release of inflammatory mediators. Curcumin’s inhibition of NF-κB signaling mitigates these deleterious processes, thereby protecting neuronal integrity. Thus, NF-κB inhibition represents a critical mechanism underlying curcumin’s neuroprotective efficacy.

#### 4.2.1. Inhibition of the NF-κB Signaling Pathway

NF-κB is a master transcription factor that orchestrates the expression of numerous pro-inflammatory genes, including IL-1β, TNF-α, IL-6, iNOS, and COX-2. Under basal conditions, NF-κB is sequestered in the cytoplasm as an inactive complex with the inhibitor protein IκBα. Upon stimulation by agents such as lipopolysaccharide (LPS), IκB kinase (IKK) phosphorylates IκBα, triggering its ubiquitination and proteasomal degradation. This degradation liberates NF-κB, allowing its nuclear translocation and subsequent initiation of inflammatory gene transcription [71].

Curcumin modulates this signaling cascade at multiple checkpoints. It inhibits IKKβ activity, thereby preventing IκBα phosphorylation and degradation. Additionally, curcumin stabilizes IκBα, preserving the NF-κB/IκBα complex in the cytoplasm and inhibiting NF-κB nuclear translocation. Furthermore, curcumin attenuates the DNA-binding activity of NF-κB, cumulatively resulting in suppressed transcription of pro-inflammatory target genes. These multi-level inhibitory effects effectively reduce inflammation in both peripheral tissues and the central nervous system.

#### 4.2.2. Inhibition of COX-2 Expression

COX-2 is an inducible enzyme predominantly upregulated in response to inflammatory stimuli and is critical for the biosynthesis of pro-inflammatory prostaglandins, notably prostaglandin E_2_ (PGE_2_). Curcumin exerts dose-dependent inhibition of COX-2 mRNA and protein expression across various experimental inflammation models, mainly through the suppression of transcription factors NF-κB and activator protein 1 (AP-1). This suppression is particularly relevant in neuroinflammation, where elevated PGE_2_ levels contribute to microglial activation and exacerbate neuronal damage [72,73]. For instance, in LPS-stimulated murine models, curcumin administration at 50–100 mg/kg significantly reduced COX-2 expression in brain tissue, correlating with attenuated neuroinflammatory responses.

#### 4.2.3. Downregulation of iNOS Expression and Nitric Oxide Production

Inducible nitric oxide synthase (iNOS) is a key inflammatory enzyme responsible for high-output production of nitric oxide under pathological conditions. Excessive NO, particularly when reacting with superoxide anion (O_2_^−^) to generate peroxynitrite (ONOO^−^), exerts cytotoxic effects through mechanisms such as DNA damage, lipid peroxidation, and protein nitration. Curcumin effectively suppresses iNOS gene transcription and protein expression, leading to a significant reduction in NO production. These inhibitory effects have been extensively validated in multiple in vitro models, including lipopolysaccharide (LPS)-stimulated BV2 microglial cultures, as well as in in vivo models of inflammation [74,75,76,77,78,79]. For example, curcumin at concentrations of 10–20 µM markedly decreased iNOS expression and NO release in LPS-treated BV2 cells. Similarly, oral administration of curcumin at 100 mg/kg in murine models of colitis resulted in reduced intestinal iNOS expression, which correlated with improved histopathological outcomes [80].

### 4.3. Role of Curcumin in Apoptosis Regulation: Molecular Mechanisms and Neuroprotective Relevance

Programmed cell death (apoptosis) plays a central role in the pathophysiology of neurodegenerative diseases. Oxidative stress, inflammatory mediators, mitochondrial dysfunction, and DNA damage contribute to pathological neuronal loss, particularly in Alzheimer’s disease, Parkinson’s disease, and ischemic brain injury. Curcumin has been demonstrated to modulate major apoptotic signaling pathways, thereby promoting neuronal survival and preservation of neural tissue.

#### 4.3.1. Inhibition of Intrinsic (Mitochondrial) Apoptosis

The intrinsic apoptotic pathway is predominantly activated by oxidative stress or mitochondrial dysfunction, with the B-cell lymphoma 2 (Bcl-2) protein family serving as key regulators. Curcumin has been reported to downregulate the expression of pro-apoptotic Bax, while upregulating anti-apoptotic proteins such as Bcl-2 and myeloid cell leukemia 1 (Mcl-1). This shift in the Bcl-2/Bax ratio prevents mitochondrial outer membrane permeabilization (MOMP), thereby inhibiting the release of cytochrome c into the cytosol and subsequent activation of the apoptosome and caspase-9—caspase-3 cascade. This mechanism is especially relevant in neurons, where curcumin selectively targets pathologically altered or dysfunctional cells for apoptosis while exerting minimal cytotoxicity on healthy neuronal populations. Such selective modulation underscores curcumin’s therapeutic potential in neurodegenerative disorders [81,82,83,84,85,86].

#### 4.3.2. Modulation of Extrinsic, Death Receptor-Mediated Apoptosis

Curcumin also modulates the extrinsic apoptotic pathway, which is triggered by activation of death receptors on the cell surface, such as Fas (CD95) and tumor necrosis factor-related apoptosis-inducing ligand receptor (TRAIL-R). Activation of these receptors leads to recruitment of adaptor proteins and activation of initiator caspase-8, which directly cleaves and activates executioner caspase-3. Alternatively, caspase-8 cleaves BH3-interacting domain death agonist (Bid), linking to the intrinsic mitochondrial apoptotic pathway.

Preclinical studies have shown that curcumin enhances the cleavage of caspase-8 and caspase-3, suggesting a dual mechanism promoting apoptosis. This may facilitate selective elimination of inflamed, damaged, or dysfunctional cells within the CNS, contributing to the restoration of tissue homeostasis in neurodegenerative conditions [87,88,89,90].

#### 4.3.3. Modulation of Apoptosis-Related Signaling Pathways: NF-κB, p53, and MAPK

Curcumin exerts complex regulatory effects on key apoptosis-related signaling pathways, including nuclear factor kappa B (NF-κB), tumor protein p53, and mitogen-activated protein kinases (MAPKs) [91,92,93].

NF-κB pathway: Curcumin inhibits NF-κB activation by stabilizing its inhibitor IκBα, preventing NF-κB nuclear translocation. This results in decreased transcription of anti-apoptotic genes such as Bcl-2, Mcl-1, and X-linked inhibitor of apoptosis protein (XIAP). In the context of chronic neuroinflammation, this may restore apoptosis susceptibility in damaged neurons.p53 pathway: Curcumin upregulates p53, a key mediator of the cellular stress response. Activated p53 enhances transcription of pro-apoptotic genes such as Bax, promoting mitochondrial-mediated apoptosis in response to oxidative stress or DNA damage, thereby supporting neuronal quality control.MAPK pathway: Curcumin modulates multiple MAPK signaling branches. It inhibits p38 MAPK, implicated in inflammation and apoptosis, and regulates c-Jun *N*-terminal kinase (JNK) and extracellular signal-regulated kinase (ERK) pathways, exerting context-dependent effects on neuronal survival and stress response.

#### 4.3.4. Bax/Bcl-2 Ratio as a Molecular Marker

Curcumin’s modulation of the Bax/Bcl-2 ratio is well documented and serves as a reliable indicator of intrinsic apoptosis activation. An increased Bax/Bcl-2 ratio correlates with mitochondrial membrane permeabilization and progression toward apoptosis. Numerous in vitro and in vivo studies have confirmed the relevance of this parameter in assessing curcumin’s neuroprotective efficacy [94].

### 4.4. Modulation of Pathological Protein Aggregation

Curcumin exerts neuroprotective effects, in part, by modulating the aggregation of key pathological proteins implicated in neurodegenerative disorders, including amyloid-β (Aβ) in Alzheimer’s disease, α-synuclein in Parkinson’s disease, and Tau in tauopathies.

#### 4.4.1. Amyloid-β Aggregation

Among curcumin’s diverse neuroprotective properties, its capacity to inhibit Aβ aggregation remains particularly significant in AD pathogenesis. The progression of AD is closely linked to misfolding and self-assembly of Aβ peptides into soluble oligomers and insoluble fibrils, which contribute to synaptic dysfunction, neuronal toxicity, and extracellular plaque deposition.

Curcumin has been shown to bind directly to Aβ oligomers and fibrils, stabilizing them in more soluble and less neurotoxic conformations, thereby inhibiting further fibrillogenesis [95,96]. This anti-aggregation activity is primarily attributed to curcumin’s molecular structure—especially its aromatic rings and β-diketone moiety—which facilitate π–π stacking interactions, hydrophobic contacts, and hydrogen bonding with Aβ residues. Structure–activity relationship studies revealed that curcumin derivatives bearing hydroxyl or trifluoromethoxy substituents enhance inhibition of Aβ fibril formation [97,98]. Furthermore, curcumin chelates redox-active metal ions such as Cu^2+^, Zn^2+^, and Fe^3+^, which are known to accelerate Aβ aggregation and promote ROS generation [99,100].

#### 4.4.2. α-Synuclein Aggregation

α-Synuclein aggregation is a central pathological mechanism in PD, leading to the formation of Lewy bodies and neurodegeneration [101]. Curcumin has been reported to inhibit α-synuclein fibrillation and stabilize non-toxic oligomeric species, reducing cytotoxicity in cellular and animal PD models [102,103]. This effect appears to involve direct binding to α-synuclein monomers and oligomers, modulation of hydrophobic interactions, and interference with β-sheet formation, positioning curcumin as a potential disease-modifying agent in synucleinopathies.

#### 4.4.3. Tau Aggregation and Hyperphosphorylation

Hyperphosphorylation of the Tau protein contributes to neurofibrillary tangle formation in AD and other tauopathies [104]. Preclinical and clinical evidence suggest that curcumin can interfere with Tau hyperphosphorylation and aggregation [105,106,107]. Notably, a Longvida^®^ study demonstrated reductions in Tau dimer levels following curcumin supplementation [106,108,109]. Mechanistically, curcumin may modulate Tau kinases and phosphatases, reduce oxidative stress, and inhibit inflammatory pathways that promote Tau pathology [107]. Through these complementary mechanisms—including direct interactions with Aβ, α-synuclein, and Tau, inhibition of fibrillogenesis, and metal ion chelation—curcumin exhibits broad anti-protein aggregation activity. This multi-targeted mode of action supports its potential as an adjunctive or preventive therapeutic strategy across neurodegenerative disorders characterized by pathological protein aggregation.

### 4.5. Regulation of Autophagy and Mitophagy

Curcumin exerts neuroprotective effects partly through the modulation of autophagy and mitophagy—two fundamental cellular quality control mechanisms essential for maintaining neuronal homeostasis. Autophagy is a highly conserved lysosome-dependent catabolic process responsible for the degradation of damaged organelles, misfolded proteins, and cytoplasmic debris. In post-mitotic neurons, which lack proliferative capacity, efficient autophagic activity is crucial for proteostasis and the prevention of neurodegenerative pathology.

A substantial body of evidence indicates that curcumin enhances autophagic flux, as demonstrated by increased expression of key autophagy markers such as LC3-II (microtubule-associated protein 1A/1B-light chain 3, form II) and Beclin-1, which are involved in autophagosome formation and initiation, respectively. Curcumin promotes the conversion (lipidation) of LC3-I to LC3-II and upregulates Beclin-1 expression, reflecting the activation of the autophagic machinery [110,111].

Moreover, curcumin facilitates mitophagy, the selective degradation of damaged or depolarized mitochondria via autophagy, thus preserving mitochondrial integrity and cellular bioenergetics. It activates key mitophagy regulators, including PTEN-induced kinase 1 (PINK1) and the E3 ubiquitin ligase Parkin, which coordinate the ubiquitination and lysosomal targeting of dysfunctional mitochondria [112]. This enhancement of mitophagy contributes to increased cellular resilience against stress, mitigates apoptotic signaling pathways, and supports neuronal survival. Importantly, curcumin’s ability to induce autophagy and mitophagy extends beyond neuroprotection, bearing implications in oncology. By promoting the clearance of toxic protein aggregates and damaged mitochondria, curcumin sensitizes cancer cells to chemotherapeutic agents and disrupts tumor cell survival mechanisms [113].

### 4.6. Induction of Neurotrophic Factors

Brain-derived neurotrophic factor (BDNF) is a crucial neurotrophin in the central nervous system, playing a vital role in neuronal survival, differentiation, synaptogenesis, long-term potentiation (LTP), and overall neuroplasticity [114,115,116,117,118,119]. BDNF is abundantly expressed in brain regions integral to cognitive and emotional processing, including the hippocampus, prefrontal cortex, and amygdala [120].

Numerous studies have demonstrated that systemic administration of curcumin significantly upregulates both BDNF mRNA and protein expression, particularly within the hippocampus. This modulation likely underpins the cognitive enhancement and neuroprotective effects observed in preclinical models [121]. The upregulation of BDNF is mediated through activation of intracellular signaling pathways, notably the extracellular signal-regulated kinase/mitogen-activated protein kinase (ERK/MAPK) cascade and the cAMP response element-binding protein (CREB), both well-established regulators of BDNF gene transcription [11].

Elevated BDNF levels enhance synaptic plasticity and promote dendritic spine formation, processes essential for memory consolidation and learning. Correspondingly, curcumin treatment is associated with improved cognitive performance and memory restoration in models of neurodegeneration and chronic stress [122].

Beyond Alzheimer’s disease, increased BDNF expression may also confer neuroprotective effects in Parkinson’s disease by supporting the survival and function of dopaminergic neurons in the substantia nigra pars compacta. Curcumin-induced elevation of BDNF in this context may contribute to stabilization of dopaminergic neurotransmission and amelioration of motor deficits [11]. In summary, induction of BDNF constitutes a pivotal mechanism underlying curcumin’s neuroprotective actions, reinforcing its therapeutic potential as an adjunctive or preventive strategy in Alzheimer’s disease, Parkinson’s disease, and other CNS disorders.

### 4.7. Putative Cerebrovascular Effects of Curcumin and Their Contribution to Neuroprotection

Cerebrovascular health is a critical determinant of brain function, particularly in aging, where structural and functional alterations in the microvasculature contribute substantially to cognitive decline and neurodegeneration [123,124,125,126,127,128,129,130,131,132,133,134,135,136,137,138,139,140]. Age-related changes in cerebral microcirculation include impaired neurovascular coupling (NVC) [124,127,129,132,134,136,137,138,139,141,142,143,144,145,146,147,148,149,150], blood–brain barrier disruption [123,124,127,128,129,131,135,136,141,146,151,152,153,154,155,156,157,158,159,160,161,162,163,164,165,166], dysregulation of cerebral blood flow (CBF), endothelial dysfunction [141,167], endothelial senescence [168,169,170,171,172,173,174,175,176], microvascular rarefaction [128,177], and chronic low-grade microvascular inflammation [164,178,179]. These pathological processes compromise neuronal homeostasis by impairing oxygen and nutrient delivery, increasing exposure to neurotoxic blood-derived molecules, and disrupting the tightly regulated cellular milieu necessary for synaptic function and neuronal survival. Importantly, these microvascular disturbances are now recognized as major contributors not only to vascular cognitive impairment and dementia [180] but also to AD, PD [181,182], and post-stroke cognitive impairment [183,184,185,186,187,188,189,190,191,192], where mixed vascular–neurodegenerative pathology is common [127,129,193,194].

Curcumin’s neuroprotective effects may be partly mediated through its capacity to preserve and restore cerebrovascular health. A growing body of preclinical and in vitro evidence indicates that curcumin exerts a broad range of vasoprotective actions that are particularly relevant in the context of age-related microvascular alterations. NVC impairment—a hallmark of cerebrovascular aging—reduces the brain’s ability to match local blood flow to neuronal activity, thereby limiting oxygen and nutrient delivery to metabolically active regions [124,141,195]. By virtue of its potent anti-inflammatory and antioxidant effects, particularly through inhibition of NF-κB [196] and activation of Nrf2 signaling [197,198,199,200], curcumin can restore endothelial nitric oxide synthase (eNOS) activity, normalize vascular tone, and improve stimulus-induced cerebral blood flow responses. Another critical target is the BBB, which becomes increasingly vulnerable to disruption with age and in neurodegenerative diseases, leading to the extravasation of plasma proteins, infiltration of immune cells, and amplification of neuroinflammation [127,136,157,160]. Curcumin has been shown to upregulate tight junction proteins such as claudin-5, occludin, and ZO-1, while suppressing matrix metalloproteinase-9 (MMP-9), a key mediator of BBB breakdown [201]. Through these actions, curcumin reduces vascular permeability and protects the delicate microenvironment of the brain parenchyma.

Age-related endothelial dysfunction and reduced nitric oxide bioavailability also contribute to cerebral hypoperfusion, a well-recognized risk factor for neurodegeneration. Curcumin’s ability to scavenge reactive oxygen species and suppress vascular NADPH oxidase activity helps preserve nitric oxide signaling, thereby supporting endothelium-dependent vasodilation and stabilizing cerebral blood flow regulation. In addition, curcumin mitigates endothelial senescence—a process that drives a pro-inflammatory, pro-thrombotic, and vasoconstrictive endothelial phenotype—by reducing oxidative DNA damage, suppressing the senescence-associated secretory phenotype (SASP), and promoting endothelial cell survival and proliferation, ultimately maintaining microvascular integrity in the aging brain.

Furthermore, curcumin may counteract microvascular rarefaction, the age-related reduction in capillary density that limits metabolic support to neurons and increases their vulnerability to injury. By modulating angiogenic pathways, particularly through vascular endothelial growth factor (VEGF) and phosphoinositide 3-kinase/protein kinase B (PI3K/Akt) signaling, curcumin supports the maintenance of a functional capillary network. Furthermore, its suppression of chronic, low-grade microvascular inflammation—achieved through inhibition of pro-inflammatory cytokines such as interleukin-1 beta (IL-1β), tumor necrosis factor-alpha (TNF-α), and interleukin-6 (IL-6), as well as downregulation of adhesion molecules including intercellular adhesion molecule-1 (ICAM-1) and vascular cell adhesion molecule-1 (VCAM-1)—reduces leukocyte–endothelial interactions and prevents secondary vascular injury.

Taken together, these vasoprotective effects complement curcumin’s direct neuroprotective actions, such as inhibition of amyloid aggregation, enhancement of autophagy, and upregulation of neurotrophic factors. In age-related central nervous system disorders, the preservation of microvascular function may therefore represent a crucial mechanistic link between curcumin’s systemic vascular benefits and its ability to promote neuronal resilience and cognitive health.

## 5. Preclinical Evidence

Curcumin has garnered significant scientific interest as a potential adjunct therapy for various neurological and psychiatric disorders [202]. Extensive preclinical research has demonstrated its robust anti-inflammatory, antioxidant, anti-apoptotic, and neuroprotective properties across diverse models of nervous system diseases, including Alzheimer’s disease, Parkinson’s disease, depression, anxiety, stroke, and autism spectrum disorders.

These neuroprotective effects are mediated by multiple mechanisms, such as the downregulation of pro-inflammatory cytokines (e.g., TNF-α, IL-1β), reduction of reactive oxygen species (ROS), and inhibition of apoptosis through modulation of the Bax/Bcl-2 ratio. Additionally, curcumin suppresses pathological protein aggregation, notably amyloid-β and α-synuclein, which are pivotal in the pathogenesis of neurodegenerative diseases [15,203,204,205]. The following section offers a comprehensive overview of preclinical findings, with a focus on Alzheimer’s disease, Parkinson’s disease, and ischemic stroke models.

### 5.1. Animal Model Outcomes

#### 5.1.1. Alzheimer’s Disease

Numerous studies have demonstrated curcumin’s neuroprotective effects in various animal models of Alzheimer’s disease. Sun et al. [45] reported that oral administration of curcumin in amyloid precursor protein/presenilin-1 (APP/PS1) transgenic mice improved spatial memory and reduced hippocampal amyloid-beta (Aβ) plaque deposition, effects linked to modulation of gut microbiota composition. Lin et al. [206] found that combined treatment with curcumin and berberine decreased Aβ_1–42_ levels, neuroinflammation, and oxidative stress, thereby enhancing cognition through modulation of amyloid precursor protein processing and activation of the AMPK–autophagy pathway. Similarly, Su et al. [207] showed that curcumin improved learning performance and reduced Aβ burden while attenuating microglial activation; these effects were attributed to enhanced bioavailability and regulation of NF-κB, mTOR, and Nrf2 signaling pathways.

In aluminum chloride-induced AD rat models, curcumin ameliorated cognitive deficits, reduced anxiety-like behavior, restored cholinergic function, and mitigated oxidative stress and inflammation [208]. In vitro, curcumin reversed Aβ-induced mitochondrial dysfunction and promoted mitochondrial biogenesis and synaptic protein preservation [209]. Wang et al. [210] demonstrated that curcumin suppressed astrocyte activation and neuroinflammation, resulting in improved memory in Aβ_40_-induced rats.

Nanocurcumin formulations have shown improved brain bioavailability, reduced amyloid plaque load, and enhanced cognitive performance in transgenic mice [211]. Likewise, NanoCurc™ decreased oxidative stress and apoptosis while augmenting antioxidant defenses both in vitro and in vivo [212].

Begum et al. [213] and Xiong et al. [214] reported that curcumin and its metabolite tetrahydrocurcumin reduced neuroinflammation and oxidative damage; notably, only curcumin prevented amyloid plaque deposition and protein oxidation, underscoring the essential role of its dienone bridge in anti-amyloidogenic activity.

Additional preclinical studies have further confirmed curcumin’s neuroprotective effects across diverse AD models. In intracerebroventricular streptozotocin (ICV-STZ)-induced rats, curcumin improved memory performance, attenuated oxidative stress, and enhanced cholinergic activity [215]. In D-galactose-induced aged mice, curcumin promoted hippocampal neurogenesis and increased brain-derived neurotrophic factor levels, leading to improved cognitive function [216]. Moreover, in behavioral assays including the object recognition test (ORT), object location test (OLT), and Y-maze, administration of curcumin at doses ranging from 50 to 100 mg/kg enhanced recognition memory, accompanied by modest anti-inflammatory effects, although no significant changes in neurogenesis were observed [217].

In human Tau (hTau) transgenic mice, curcumin formulated as Longvida significantly reduced Tau dimer levels, enhanced synaptic function, and upregulated the expression of heat shock proteins 70 and 90 (HSP70/90) [108]. Aerosolized curcumin administration decreased amyloid-beta plaque burden and improved spatial memory performance in 5XFAD transgenic mice [218]. In vitro studies under Aβ-induced oxidative stress conditions demonstrated that curcumin reduced lipid peroxidation, increased activities of antioxidant enzymes, and elevated levels of nerve growth factor (NGF) [219]. In amyloid precursor protein Swedish mutation (APPSw) transgenic mice, dietary curcumin treatment dose-dependently decreased interleukin-1 beta (IL-1β), oxidative stress markers, amyloid plaque accumulation, and microglial activation [220]. Notably, nanoformulated curcumin showed superior efficacy in attenuating neuroinflammation and enhancing cognitive function compared to conventional curcumin in streptozotocin (STZ)-induced Alzheimer’s disease models [221]. A summary of these neuroprotective effects is provided in Table 1.

#### 5.1.2. Parkinson’s Disease

Numerous preclinical studies highlight the neuroprotective potential of curcumin across various animal and cellular models of Parkinson’s disease. Curcumin exerts its effects through multiple mechanisms, including attenuation of oxidative stress and inflammation, mitochondrial protection, and modulation of autophagy and protein aggregation.

In a copper-induced parkinsonism model, Abbaoui et al. [222] demonstrated that curcumin restored tyrosine hydroxylase (TH) expression in dopaminergic brain regions and improved motor function, likely through its antioxidant and anti-inflammatory properties. Similarly, Rajeswari et al. [223] reported that curcumin and its metabolite tetrahydrocurcumin reversed reductions in dopamine and 3,4-dihydroxyphenylacetic acid (DOPAC) levels, and inhibited monoamine oxidase-B (MAO-B) activity in a 1-methyl-4-phenyl-1,2,3,6-tetrahydropyridine (MPTP) mouse model. Pan et al. [224] found that curcumin suppressed c-Jun *N*-terminal kinase (JNK) phosphorylation, thereby preventing mitochondrial apoptosis events such as Bcl-2-associated X protein (Bax) translocation and cytochrome c release, ultimately preserving dopaminergic neurons.

In a human neuroblastoma SH-SY5Y cell line overexpressing the A53T mutant α-synuclein, Jiang et al. [225] showed that curcumin restored autophagic activity by inhibiting the mechanistic target of rapamycin (mTOR)/ribosomal protein S6 kinase beta-1 (p70S6K) signaling pathway, reducing α-synuclein accumulation. Similarly, Wang et al. [226] demonstrated that curcumin decreased reactive oxygen species production and apoptosis in SH-SY5Y cells treated with α-synuclein oligomers, while stabilizing these aggregates.

In a rotenone-induced mouse model, Ramires Júnior et al. [227] reported that nanoemulsion-formulated curcumin improved motor function, reduced lipid peroxidation, preserved mitochondrial complex I activity, and enhanced endogenous antioxidant defenses. Yang et al. [228] observed that curcumin enhanced learning and memory performance and increased hippocampal dopamine and norepinephrine levels, likely via activation of the brain-derived neurotrophic factor (BDNF)/tropomyosin receptor kinase B (TrkB)/phosphoinositide 3-kinase (PI3K) signaling pathway. Fikry et al. [229] found that curcumin protected cerebellar Purkinje neurons, attenuated astrogliosis, and restored motor function following prolonged rotenone exposure.

Jagatha et al. [230] demonstrated that curcumin increased glutathione synthesis by activating γ-glutamylcysteine ligase (γ-GCL), reduced protein oxidation, and stabilized mitochondrial function both in vitro and in a glutathione-deficient mouse model. Sharma et al. [231] reported that in a lipopolysaccharide (LPS)-induced rat model, curcumin suppressed inflammatory markers (GFAP, NF-κB, TNF-α, IL-1β), improved redox balance, reduced iron accumulation, and prevented α-synuclein aggregation.

A systematic review encompassing 13 preclinical studies with 298 animals confirmed curcumin’s efficacy in reducing dopaminergic neuron loss, elevating striatal dopamine levels, and decreasing oxidative stress, inflammation, and apoptosis in the substantia nigra and striatum (*p* < 0.05) [232]. Liu et al. [233] demonstrated that high-dose curcumin (160 mg/kg/day) significantly improved motor function and preserved TH-positive dopaminergic neurons in a 6-hydroxydopamine (6-OHDA) rat model, indicating dose-dependent neuroprotection via reduction in oxidative stress and neuroinflammation. Similarly, curcumin oil solution (Curoil) enhanced motor performance and dopaminergic neuron survival in an MPTP-induced PD mouse model, likely due to improved bioavailability [234]. Collectively, these findings support curcumin’s multifaceted neuroprotective actions across key pathogenic mechanisms in PD, underscoring its potential as a candidate for future disease-modifying therapies. A summary of these effects is presented in Table 2.

#### 5.1.3. The Effects of Curcumin in Preclinical Models of Ischemic and Hemorrhagic Stroke

Extensive preclinical studies have demonstrated curcumin’s neuroprotective effects in various cerebrovascular disorder models, including ischemic and hemorrhagic stroke [39]. Despite variability in animal models, dosing, and treatment regimens, these studies consistently highlight curcumin’s potent anti-inflammatory, antioxidant, and anti-apoptotic properties, along with its ability to modulate microglial polarization.

In a middle cerebral artery occlusion (MCAO) model, Li et al. [235], showed that co-administration of 100 mg/kg curcumin with human umbilical cord mesenchymal stem cells (hUC-MSCs) significantly reduced brain edema and infarct volume while improving neurological function. This neuroprotection was mediated via activation of the AKT/GSK-3β/β-TrCP/Nrf2 signaling pathway, suppression of oxidative stress, and promotion of M2 microglial polarization. Jia et al. [236] reported that 300 mg/kg curcumin upregulated peroxiredoxin 6 (Prdx6) through SP1 induction, which was critical for its antioxidant and neuroprotective effects, as inhibition of Prdx6 abolished these benefits.

In a global cerebral ischemia model, Altinay et al. [237], demonstrated that combined pre- and post-treatment with curcumin enhanced antioxidant defenses (SOD, CAT, GPx), reduced oxidative and inflammatory markers (IL-6, TNF-α, MDA), and attenuated neuronal apoptosis. Marques et al. [238] showed that nanoemulsified curcumin improved motor function, reduced hematoma volume, and minimized weight loss more effectively than free curcumin in hemorrhagic stroke models, attributed to superior tissue penetration and bioavailability.

Ran et al. [239] found that 150 mg/kg curcumin mitigated white matter injury and improved sensorimotor function via inhibition of the NF-κB/NLRP3 inflammasome pathway and suppression of microglial pyroptosis. Xie et al. [240] in both in vivo and in vitro studies, reported that curcumin protected neurons by inhibiting mitochondrial apoptotic pathways—suppressing Bax activation, upregulating Bcl-2, decreasing caspase-3 activity, and preserving mitochondrial membrane potential.

Li et al. [241] showed that curcumin (300 mg/kg, administered 30 min before reperfusion) significantly reduced infarct size and brain edema and improved neurological scores, effects linked to decreased NF-κB, ICAM-1, MMP-9, and caspase-3 expression, suggesting preservation of blood–brain barrier (BBB) integrity. Liu et al. [242] demonstrated that 7-day curcumin treatment (300 mg/kg) promoted neurogenesis and improved behavioral outcomes through Notch pathway activation, evidenced by increased BrdU^+^ and BrdU/DCX^+^ cell counts. Wu et al. [243] reported that curcumin pretreatment upregulated tight junction proteins (ZO-1, occludin, claudin-5) and inhibited NF-κB and MMP-9, reducing BBB permeability and inflammation.

In a distal MCAO model, Liu et al. [244], found that curcumin (150 mg/kg, twice intraperitoneally) decreased infarct volume, enhanced M2 microglial activation (CD206^+^Iba1^+^), suppressed M1 phenotype (CD16^+^Iba1^+^), and lowered pro-inflammatory cytokines (TNF-α, IL-6, IL-12p70). Wu et al. [245] confirmed that curcumin reduced neuronal injury and oxidative stress in oxygen-glucose deprivation/reperfusion (OGD/R) and MCAO models by activating the AKT/Nrf2 signaling axis. Jiang et al. [246] demonstrated that intravenous curcumin protected against BBB disruption, brain edema, and neuroinflammation by inhibiting the iNOS/NO pathway in MCAO rats. Zhang et al. [247] further validated curcumin’s anti-inflammatory and anti-apoptotic actions by reducing TNF-α and p53, and increasing Sirt1 and Bcl-2 expression.

Collectively, these studies (summarized in Table 3) provide robust evidence that curcumin exerts multifaceted neuroprotective effects in experimental ischemic and hemorrhagic stroke models. Future clinical trials are warranted to evaluate the translational potential of these findings for human stroke therapy.

### 5.2. Dose–Response Relationships

Preclinical animal studies have generally characterized the dose–response relationship of curcumin, revealing that moderate but detectable neuroprotective effects typically occur at doses between 25 and 50 mg/kg. At these levels, curcumin primarily enhances antioxidant activity, reduces apoptosis, and improves behavioral outcomes in models of stroke and neurodegenerative diseases. More pronounced and consistent neuroprotection is observed at higher doses, particularly ≥100 mg/kg, reflecting stronger activation of mechanisms such as oxidative stress reduction, anti-inflammatory responses, and cytoprotective signaling pathways.

Several investigations report a sigmoidal dose–response curve, with biologically significant effects becoming measurable above a threshold dose of approximately 80 mg/kg. Below this threshold, curcumin’s effects may be subtle or difficult to quantify, whereas efficacy sharply increases beyond it. This dose-dependence corresponds to curcumin’s modulation of multiple cellular pathways, including the attenuation of reactive oxygen species (ROS) production, mitigation of oxidative stress-induced damage, and regulation of key signaling cascades such as AMP-activated protein kinase (AMPK), AKT/mTOR, and NF-κB. These molecular actions contribute to improved cellular function and may delay the progression of age-related disorders, including neurodegenerative and cardiovascular diseases.

Despite curcumin’s inherently low bioavailability and rapid metabolism necessitating higher doses, toxicological studies generally support a favorable safety profile for high-dose administration. To enhance efficacy, various nanoformulations—such as nanoemulsions and nanoparticles—are increasingly employed to improve targeted delivery and bioavailability.

Future research should prioritize the development of curcumin analogs with superior pharmacokinetic properties (e.g., TML-6) and the optimization of dose–response outcomes through nanotechnology and pharmaceutical innovation. Given that most data derive from preclinical models, further well-designed human clinical trials are critical to safely translate these dose-dependent effects into clinical practice, particularly for the prevention and treatment of age-related diseases [248]. In summary, the dose–response relationship underscores that curcumin’s therapeutic efficacy is strongly dose-dependent. Identifying the optimal dosing regimen remains essential to maximize its neuroprotective and antioxidant potential, especially in the context of aging and neurodegeneration.

### 5.3. Nanoparticle-Based Formulations

Despite curcumin’s promising therapeutic potential, its clinical utility is limited by poor bioavailability, primarily due to low absorption, rapid metabolism, and systemic clearance when administered in conventional forms. To overcome these pharmacokinetic challenges, a variety of nanotechnology-based delivery systems have been developed, substantially enhancing curcumin’s bioavailability and efficacy. Notably, encapsulation within liposomes, poly(lactic-co-glycolic acid) (PLGA) nanoparticles, and saponin-coated nanocarriers has been shown to increase bioavailability up to nine-fold compared to traditional curcumin formulations, even when co-administered with bioenhancers such as piperine [249].

Key advantages of these nanoformulations include:Improved brain targeting: Nanoparticles facilitate curcumin’s transport across the blood–brain barrier, ensuring more effective delivery to brain tissues. This capability is crucial for therapeutic applications in neurodegenerative diseases, stroke, and related disorders.Reduced clearance: Nanoformulations prolong systemic circulation time by slowing curcumin’s elimination, thereby increasing its bioactive presence and therapeutic impact.Enhanced neuroprotective effects: Preclinical studies demonstrate that PLGA- or liposome-encapsulated curcumin can achieve similar or superior neuroprotective outcomes at lower doses than free curcumin, improving treatment consistency and reproducibility.

Moreover, liposomal hydrogels have been engineered for injectable, localized, and controlled release of curcumin, such as in promoting wound healing post-tumor resection. Clinical investigations in humans further confirm that nanocurcumin formulations yield superior therapeutic effects compared to conventional preparations, owing to more rapid absorption and improved tissue accumulation [249,250,251].

Overall, nanoparticle-based curcumin formulations represent a promising strategy to circumvent bioavailability limitations, enabling targeted and efficacious interventions for age-related diseases, neurodegenerative disorders, and cancer. Continued clinical research is essential to identify optimal formulations and administration routes to maximize therapeutic benefits.

## 6. Neurological Effects of Curcumin: Clinical Outcomes

Clinical trials investigating curcumin’s neurological effects have yielded variable results, though encouraging trends have emerged across several domains. In the management of depression, curcumin has demonstrated efficacy particularly in mild to moderate cases, likely via modulation of monoaminergic neurotransmission and inflammatory pathways [203]. For neuropathic and postoperative pain, diverse formulations—including nanocurcumin—have shown analgesic effects by attenuating inflammation and oxidative stress [204]. Regarding cognitive decline, Alzheimer’s disease, and aging-related impairments, multiple studies report that curcumin supplementation improves memory, attention, and learning, especially with long-term use of highly bioavailable formulations [14,15,205,252]. Although clinical data on stroke remain limited, preclinical models suggest curcumin may reduce brain injury and promote neural regeneration [205]. Additionally, both individuals with neurocognitive disorders and healthy adults have exhibited enhanced information processing, memory, and attention following curcumin intake, despite many studies involving relatively small cohorts [253]. Overall, curcumin represents a multifaceted natural compound with compelling neuroprotective potential established in preclinical research. While clinical evidence is promising, it is not yet conclusive. Future well-powered, rigorously controlled randomized trials are warranted to define optimal formulations, dosing strategies, and therapeutic indications to safely and effectively harness curcumin’s neurological benefits [203,204,254]. A detailed summary of clinical findings is presented in Table 4.

### Human Clinical Trials

Numerous human clinical trials have investigated the neurological and systemic effects of curcumin, yielding varied but promising results. Small et al. [255] conducted an 18-month study involving 40 non-demented middle-aged and older adults (51–84 years), administering Theracurmin^®^ at 90 mg twice daily (180 mg/day). Significant improvements were observed in verbal and visual memory as well as attention. PET imaging revealed reduced amyloid and tau accumulation, likely mediated by anti-inflammatory and anti-amyloid mechanisms involving the Akt/Nrf2 signaling pathway.

Cox et al. [256] evaluated the effects of 400 mg/day Longvida^®^ curcumin in healthy adults aged 60–85 over 4 weeks. Acute supplementation (1 h post-dose) improved attention and working memory, while chronic use reduced fatigue and LDL cholesterol and improved mood, attributed to antioxidant, anti-inflammatory, and lipid metabolism-modulating effects.

Baum et al. [257] administered 1 g or 4 g/day of curcumin to 34 patients with probable or possible Alzheimer’s disease over 6 months. While no significant cognitive improvements were detected via MMSE, curcuminoids were measurable in plasma, with a trend toward increased Aβ40 levels. This study underscored the low bioavailability of curcumin and better absorption from capsule formulations.

A double-blind, placebo-controlled trial with 36 patients [258], using Curcumin C3 Complex^®^ at 2 g and 4 g/day for 24 weeks reported no significant adverse events. Although no cognitive benefits were observed (ADAS-Cog), the study reinforced curcumin’s anti-inflammatory and antioxidant properties, as well as its capacity to inhibit amyloid-beta aggregation.

Conversely, a larger trial of 96 older adults [259] showed that daily supplementation with 1500 mg Biocurcumax™ (BCM-95^®^CG) for 12 months prevented cognitive decline measured by Montreal Cognitive Assessment (MoCA) scores compared to placebo. However, other cognitive metrics showed no significant difference. The authors highlighted curcumin’s antioxidant and anti-inflammatory effects and called for further biomarker research.

In Parkinson’s disease, a 9-month study with 60 patients [260], administering 80 mg nanomicellar curcumin daily found no significant improvement in motor symptoms (MDS-UPDRS) or quality of life (PDQ-39) compared to placebo, despite a trend toward improvement in the MDS-UPDRS Part III subscore. Some participants reported gastrointestinal side effects, including nausea and reflux.

In contrast, a 3-month trial with 50 PD patients [261], administering 160 mg/day nanomicellar curcumin observed significant improvements in sleep quality and overall quality of life (PDQ-39), although fatigue remained unchanged. These effects were likely mediated by antioxidant, anti-inflammatory, and neuroprotective mechanisms.

A 12-month study [262], involving 19 PD patients treated with 2 g/day of a curcumin–phospholipid complex (Meriva^®^) showed reductions in autonomic dysfunction (COMPASS-31) and non-motor symptoms (NMSS), with slower clinical progression (MDS-UPDRS, Hoehn and Yahr staging). Skin biopsies revealed decreased phosphorylated alpha-synuclein deposition, supporting curcumin’s blood–brain barrier penetration and anti-amyloidogenic, anti-inflammatory effects.

Lastly, a study of 56 post-stroke rehabilitation patients [263], supplementing with 500 mg curcumin plus 5 mg piperine daily for 12 weeks demonstrated significant reductions in hs-CRP, total cholesterol, triglycerides, carotid intima-media thickness (CIMT), body weight, waist circumference, and blood pressure, alongside increased total antioxidant capacity (TAC). Patients receiving curcumin also reported less pain progression relative to placebo. A detailed overview of these clinical trials in Alzheimer’s disease, Parkinson’s disease, and stroke is provided in Table 4.

## 7. Neurological Effects of Curcumin: A Critical Synthesis of Clinical Evidence

The outcomes of human clinical trials investigating the neuroprotective potential of curcumin are heterogeneous. Nevertheless, several studies suggest that beneficial effects may be observed under specific conditions. The aim of this synthesis is to critically examine the main factors influencing trial outcomes.

### 7.1. The Role of Bioavailability

Trials reporting negative or non-significant findings predominantly employed conventional, poorly bioavailable formulations, such as standard curcumin powder or Curcumin C3 Complex^®^ [257,258]. In these studies, plasma concentrations remained low, thereby limiting the potential for measurable clinical effects.

In contrast, trials reporting beneficial outcomes generally used advanced formulations designed to enhance absorption, including Theracurmin^®^ [255], Longvida^®^ [256], Biocurcumax™ [259], and Meriva^®^ [262]. These findings underscore bioavailability as a critical determinant of clinical efficacy in curcumin trials.

### 7.2. Characteristics of the Study Population

The characteristics of the study population also strongly influenced outcomes. Studies involving cognitively healthy participants or those with mild cognitive impairment and early-stage disease [255,256,259] were more likely to report favorable effects, such as improvements in memory, attention, and mood.

By contrast, trials conducted in patients with moderate Alzheimer’s disease [257,258] did not demonstrate consistent cognitive improvements, even at high doses. This suggests that curcumin’s primary value may lie in prevention or early intervention, while its therapeutic potential in advanced neurodegenerative stages appears limited.

### 7.3. Sensitivity of Endpoints and Biomarkers

Methodological differences across trials further contribute to inconsistent findings. Studies employing more sensitive cognitive tests and biomarker-based endpoints—such as PET imaging to assess amyloid and tau deposition [255] or skin biopsies to detect α-synuclein aggregates [262] were more likely to report beneficial effects.

In contrast, studies relying solely on conventional cognitive scales such as the MMSE or ADAS-Cog [257,258] often failed to detect significant changes. This suggests that curcumin’s effects may be more readily captured using sensitive and targeted outcome measures. Overall, current clinical evidence suggests that curcumin possesses neuroprotective potential, though the supporting data remain preliminary. The most consistent benefits are observed when:formulations with enhanced bioavailability are used [255,256,259,262]interventions are initiated in early or preclinical stages [255,256,259]and sensitive cognitive or biomarker-based endpoints are employed [255,262]

Nonetheless, the overall level of evidence remains low, and methodological heterogeneity across studies significantly limits generalizability. Future research should prioritize larger, longer-duration, randomized, and well-controlled clinical trials to establish the optimal formulation, dosage, and target population for curcumin in the prevention and treatment of neurodegenerative diseases.

## 8. Efficacy and Safety

Curcumin has emerged as a promising neuroprotective agent against neurodegenerative disorders such as Alzheimer’s disease, Parkinson’s disease, and stroke. It exerts multifaceted effects by modulating key cellular and molecular signaling pathways involved in neuronal damage, including amyloid-β aggregation, oxidative stress, and neuroinflammation. Curcumin primarily enhances antioxidant defenses and suppresses inflammatory responses through activation of the nuclear factor erythroid 2–related factor 2 (Nrf2) pathway and inhibition of NF-κB, which underlie many of its neuroprotective properties. Additionally, curcumin promotes neuronal health and function by stimulating autophagy and neurotrophic factor production. Clinical studies support curcumin’s capacity to cross the blood–brain barrier, which is critical for its central nervous system efficacy.

Despite its potent neuroprotective effects and generally low toxicity, curcumin’s clinical translation is limited by poor bioavailability, characterized by low absorption, rapid metabolism, and systemic elimination. Nanotechnology-based delivery strategies, including liposomal and nanoparticle formulations, as well as curcumin structural analogs, are emerging approaches to enhance solubility, sustain release, and improve targeted brain delivery. These nanoformulations represent a promising avenue to overcome pharmacokinetic limitations and optimize therapeutic outcomes.

Curcumin is generally well tolerated, with low toxicity and minimal adverse effects reported in conventional formulations. However, high-bioavailability formulations have occasionally been associated with liver toxicity. Reported adverse effects include nausea, vomiting, gastrointestinal discomfort, and, rarely, hepatotoxicity. Use of curcumin during pregnancy or breastfeeding may not be safe, and further studies are needed to confirm its safety in these populations.

Nonetheless, further investigations are warranted to confirm long-term safety, optimal dosing, and efficacy, particularly in combination therapies or nanoformulated systems. The precise molecular mechanisms underlying curcumin’s neuroprotective actions also require further elucidation. Collectively, these considerations highlight the need for well-designed clinical trials and continued research into nano-based delivery strategies to fully harness curcumin’s therapeutic potential in neurodegenerative disease contexts [255,256,257].

## 9. Bioavailability, Standardization

Despite its considerable therapeutic potential, curcumin’s clinical translation is significantly limited by its poor bioavailability, characterized by low oral absorption, rapid metabolism, and systemic elimination. These pharmacokinetic challenges reduce curcumin’s effective concentration in target tissues and limit its clinical efficacy, particularly in neurodegenerative and metabolic disorders. To address these limitations, a variety of advanced delivery strategies and novel formulations have been developed. Nanotechnology-based approaches—including liposomal curcumin, solid lipid nanoparticles (SLNP), polymeric nanoparticles, nanomicelles, dendrimers, nanoemulsions, and nanocrystals—have emerged as particularly promising due to their ability to enhance solubility, protect curcumin from rapid metabolism, facilitate sustained release, and improve targeted tissue and blood–brain barrier delivery [264].

Clinical and preclinical evidence supports the pharmacological benefits of these formulations [265,266,267]. Liposomal curcumin has been shown to maintain sustained plasma concentrations and elicited favorable tumor marker responses in metastatic cancer patients, indicating its potential for targeted delivery. Nanomicelle formulations improved glycemic control, lipid profiles, and inflammatory markers in diabetic patients undergoing hemodialysis, while also demonstrating bone-strengthening effects in postmenopausal women. Solid lipid curcumin particles (SLCP, e.g., Longvida^®^) increased plasma curcumin levels and were associated with cognitive improvements in healthy older adults. Nanocurcumin formulations have also shown neuroprotective efficacy in amyotrophic lateral sclerosis patients and exhibited immunomodulatory effects in COVID-19, reducing pro-inflammatory cytokine levels and improving clinical outcomes without significant adverse effects [264,265,266,267].

Beyond nanoformulations, other innovative delivery strategies have been investigated to further enhance curcumin’s pharmacokinetic and pharmacodynamic properties. Curcumin-galactomannoside (CGM) formulations utilize natural dietary fibers to improve absorption and tissue distribution [268]. Phytosomal curcumin formulations (e.g., Curserin^®^) leverage amphipathic phospholipids to increase gastrointestinal stability and bioavailability [269]. Colloidal and amorphous formulations, such as Theracurmin^®^ and CurcuRougeTM, demonstrate superior plasma concentrations and area-under-curve metrics compared to conventional curcumin extracts, translating into improved therapeutic outcomes in musculoskeletal, metabolic, and neurodegenerative disorders [270,271].

Collectively, these data underscore that optimizing curcumin formulations is critical to overcoming its intrinsic pharmacokinetic limitations. The integration of nano- and other advanced delivery systems not only enhances bioavailability but also maximizes therapeutic efficacy across a range of disease contexts, including neurodegeneration, metabolic disorders, cardiovascular disease, cancer, and inflammatory conditions. Future research should continue to elucidate the mechanisms by which these formulations enhance curcumin’s bioavailability, determine optimal dosing regimens, evaluate long-term safety, and validate clinical efficacy in well-controlled, large-scale trials [272,273].

## 10. Future Perspectives

Curcumin has demonstrated considerable neuroprotective potential; however, its clinical translation is hindered by pharmacokinetic limitations, including poor bioavailability, rapid metabolism, and limited blood–brain barrier penetration. To address these challenges, ongoing research is focused on advanced drug delivery strategies, such as nanoparticle-based systems, liposomal formulations, and structural analogues. These approaches have shown promise in improving curcumin’s solubility, stability, and BBB permeability, thereby enhancing its therapeutic profile in neurodegenerative diseases [23]. Prodrug strategies, which involve chemical modification of curcumin to increase systemic stability and prolong circulation time, may further improve its pharmacokinetics and pharmacodynamics, resulting in more consistent therapeutic exposure at target sites [274].

Combination therapy is another promising avenue, whereby curcumin is administered alongside pharmacological agents [161,172,275,276,277,278,279,280,281,282], nutraceuticals [283], dietary interventions [284,285,286,287,288,289,290,291,292] and lifestyle modifications [293,294,295,296,297,298,299,300,301,302,303,304,305,306,307,308,309,310,311] to exploit synergistic effects. This multitarget approach can modulate several pathogenic processes simultaneously—oxidative stress, neuroinflammation, and protein misfolding—thereby potentially improving overall treatment efficacy [312]. The integration of curcumin into personalized medicine frameworks also holds potential. Biomarker-driven and genotype-based patient stratification could allow treatments to be tailored to individual pathophysiological profiles, maximizing therapeutic benefit while minimizing adverse effects, especially in heterogeneous disorders such as Alzheimer’s and Parkinson’s diseases [313,314].

Realizing these advances will require multidisciplinary research bridging nanotechnology, neuropharmacology, molecular biology, and clinical sciences. Long-term, well-designed randomized controlled trials are necessary to establish optimal formulations, dosing regimens, and safety profiles. Ultimately, embedding curcumin into combination and precision medicine strategies may substantially enhance its clinical utility in neurodegenerative disease management.

## 11. Limitations

This review is narrative in nature and was not conducted as a systematic review. Consequently, it does not claim to comprehensively cover all available literature, and relevant studies may have been inadvertently omitted. The primary aim was to summarize and interpret current evidence to highlight key mechanistic insights and potential therapeutic implications, rather than to exhaustively analyze all published data. As such, the conclusions should be interpreted with caution, taking into account the potential for selection bias and the qualitative rather than quantitative synthesis of the findings.

## 12. Conclusions

Curcumin’s neuroprotective potential is increasingly supported by both preclinical and clinical evidence, particularly in Alzheimer’s disease, Parkinson’s disease, and stroke. Preclinical models have shown that curcumin modulates multiple disease-relevant pathways, including reduction in oxidative stress, suppression of neuroinflammation, and inhibition of amyloid-β aggregation. Additional mechanisms—such as stimulation of autophagy and neurotrophic factor production—further contribute to neuronal survival, regeneration, and improved cognitive performance.

Despite this strong preclinical foundation, clinical outcomes remain variable, largely due to curcumin’s low bioavailability, rapid metabolism, and restricted BBB penetration. Advances in nanotechnology, including nanoparticle encapsulation, liposomal delivery, and prodrug development, are promising solutions to these challenges, offering improved pharmacokinetics, stability, and targeted delivery. Combination therapies, leveraging curcumin’s antioxidant and anti-inflammatory properties alongside conventional pharmacological agents, may yield synergistic benefits and improve patient outcomes.

Personalized medicine approaches, guided by biomarker profiling and genetic background, may further optimize curcumin therapy for individual patients. While the evidence base is expanding, successful translation into routine clinical practice will require continued multidisciplinary research to refine formulations, elucidate mechanisms of action, and confirm long-term safety and efficacy. As a natural, multifunctional compound with a favorable safety profile, curcumin remains a highly promising candidate for the prevention and management of neurodegenerative disorders—provided its pharmacological limitations can be effectively addressed.

## Figures and Tables

**Figure 1 nutrients-17-02884-f001:**
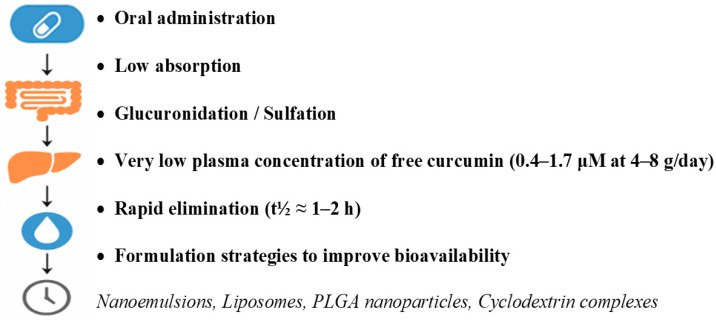
Pharmacokinetic challenges of oral curcumin.

**Figure 2 nutrients-17-02884-f002:**
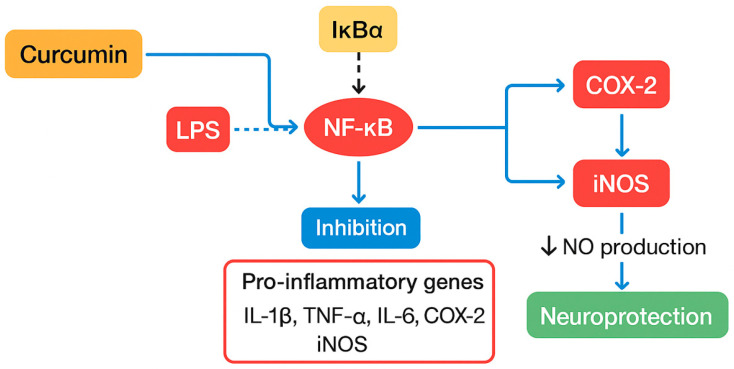
Anti-Inflammatory Mechanisms of Curcumin: Inhibition of NF-κB, COX-2, and iNOS: Curcumin reduces inflammation by inhibiting the NF-κB (nuclear factor kappa B) signaling pathway, as well as COX-2 (cyclooxygenase-2) and iNOS (inducible nitric oxide synthase) activity, thereby lowering pro-inflammatory cytokine and nitric oxide production. Arrow indicates the direction of change: ↓—decrease.

**Table 1 nutrients-17-02884-t001:** Preclinical studies on curcumin in Alzheimer’s disease models.

Ref.	Model	Dose	Duration	Route	Key Effects	Mechanisms
Sun et al. [45]	APP/PS1 male mice	50 or 200 mg/kg	3 months	Oral gavage (daily)	Improved spatial memory; reduced hippocampal Aβ plaques	Modulated gut microbiota (↑ Lactobacillaceae, ↓ Bacteroidaceae); curcumin biotransformation to neuroprotective metabolites
Lin et al. [206]	APPswe/PSEN1dE9 mice (both sexes)	Curcumin 200 mg/kg; Berberine 100 mg/kg	3 months	Oral gavage (daily)	Improved cognition; ↓ Aβ1–42, inflammation (IL-1β, TNF-α, IL-6), oxidative stress; ↑ AMPKα phosphorylation and autophagy	Synergistic anti-amyloid, anti-inflammatory, antioxidative effects; modulation of APP processing and AMPK–autophagy pathway
Su et al. [207]	Triple-transgenic Alzheimer’s disease mice	150 mg/kg (TML-6)	4 months	Oral	Improved learning; reduced brain Aβ and microglial activation	Enhanced bioavailability; reduced Aβ synthesis; ↑ ApoE; NF-κB/mTOR suppression; Nrf2 pathway activation
ELBini-Dhouib et al. [208]	AlCl_3_-induced sporadic AD rats	100 mg/kg (CUR1 co-treatment; CUR2 post-treatment)	90 days	Oral	Improved cognition; reduced anxiety; restored AChE activity; improved oxidative markers; ↓ apoptosis and inflammation	Antioxidant and anti-inflammatory actions; neuroprotection; restored cholinergic function
Reddy et al. [209]	SH-SY5Y cells + Aβ_25–35_	66.3 μM	24 h (pretreatment)	In vitro	Improved cell viability; ↑ MFN1/2, OPA1; ↓ DRP1, FIS1; ↑ PGC1α, Nrf1/2, TFAM; ↑ synaptic markers	Enhanced mitochondrial biogenesis and dynamics; synaptic protection; reduced oxidative stress
Wang et al. [210]	Aβ_1–40_-induced AD in Sprague–Dawley rats	300 mg/kg	7 days	Intraperitoneal	Improved spatial memory; reduced hippocampal GFAP mRNA; less reactive astrocytosis	Suppression of astrogliosis and neuroinflammation via GFAP downregulation
Cheng et al. [211]	Tg2576 AD mice	100 mg/kg	3 months	Oral gavage (weekly)	Improved cue memory; trend to improved working memory; ↓ amyloid plaques; ↑ brain/plasma curcumin	Nanoparticle formulation enhanced BBB penetration, pharmacokinetics, and bioavailability
Ray et al. [212]	SK-N-SH cells and athymic mice	25 mg/kg NanoCurc™ i.p. twice daily; 1 nM–5 μM in vitro	In vitro: 24 h; in vivo: ≤16 h post-dose	In vitro; i.p.	Neuroprotection from ROS; rescued injured cells; ↑ brain curcumin; ↓ ROS and caspase 3/7; ↑ GSH	Polymeric nanoparticles improved bioavailability; enhanced redox balance; antioxidant and anti-apoptotic effects
Begum et al. [213]	Tg2576 APPsw mice; LPS-injected WT mice; neuron/microglia cultures	Chronic: ~83 mg/kg diet (~500 ppm); Acute: 0.4 μmol	Chronic: ~4 months; Acute: 2 days pre-LPS	Oral (diet/gavage), i.p., i.m.	Both regimens reduced neuroinflammation (iNOS, IL-1β); Cur reduced plaques/insoluble Aβ, TC reduced soluble Aβ and JNK phosphorylation	Cur dienone bridge essential for plaque reduction; TC selectively inhibits JNK phosphorylation; both antioxidant/anti-inflammatory
Xiong et al. [214]	SH-SY5Y neuroblastoma cells (APP-transfected)	0–20 μM (24 h), or 5 μM (12–48 h)	24–48 h	In vitro	Dose- and time-dependent reduction in extracellular Aβ40/42	Inhibition of Aβ generation via PS1 and GSK-3β suppression; ↑ GSK-3β Ser9 phosphorylation; downregulation of APP-processing enzymes
Ishrat et al. [215]	ICV-STZ-infused male Wistar rats	80 mg/kg	3 weeks	Oral gavage	Improved cognition; ↓ oxidative stress (↓ 4-HNE, MDA, TBARS, H_2_O_2_, PC, GSSG; ↑ GSH, GPx, GR); ↑ choline acetyltransferase	Antioxidant activity; restored glutathione system; reduced lipid peroxidation; enhanced cholinergic function
Nam et al. [216]	Adult and D-galactose-induced aged male C57BL/6 mice	300 mg/kg Curcuma longa extract; 100 mg/kg D-galactose	10 weeks D-gal; 3 weeks C. longa	Oral gavage; subcutaneous (D-gal)	Reduced escape latency; ↑ cell proliferation (Ki67), neuroblast differentiation (DCX); ↑ pCREB, ↑ BDNF in hippocampus	CREB pathway activation; enhanced neurogenesis; increased neurotrophic factors
Bassani et al. [217]	Male Wistar rats	25, 50, or 100 mg/kg	30 days	Oral gavage	Improved object recognition (50/100 mg/kg) in ICV-STZ dementia; mild anti-inflammatory effect	Memory improvement unrelated to hippocampal neurogenesis; mild anti-inflammatory action
Ma et al. [108]	hTau transgenic mice (15–16 months) and controls	500 ppm curcumin (Longvida SLN)	4–5 months	Diet (ad libitum)	Reduced behavioral, synaptic, and chaperone deficits; ↓ soluble Tau dimers; ↑ HSP90/HSC70 in synaptic fractions	Increased HSP proteins; modulation of HSP90 client kinases; enhanced Tau clearance
McClure et al. [218]	5XFAD transgenic mice	5 mg/kg	18 weeks	Inhalation (aerosol)	Reduced Aβ plaques in hippocampus/subiculum; improved memory; no toxicity	↓ Aβ and COX-2; prevention of neuritic dystrophy and autophagic vesicle formation
Alamro et al. [219]	Primary cortical neurons (Wistar rat pups)	1 μM Aβ1–42 + 1 nM vitamin D_3_ + 5 μM curcumin	≤72 h	In vitro	Reduced lipid peroxidation; ↑ GST, catalase, SOD; improved viability; ↑ NGF	Reduced oxidative stress; enhanced antioxidant defense; increased NGF
Lim et al. [220]	APPSw Tg+ mice	160 or 5000 ppm curcumin	6 months	Diet	↓ oxidized proteins, IL-1β, Aβ plaques; ↓ GFAP (low dose); ↓ microglial activation	Anti-inflammatory (IL-1β); reduced oxidative stress and plaque formation; microglial suppression
Savall et al. [221]	STZ-induced AD male Wistar rats	6 mg/kg	14 days treatment (total 36 days)	Oral gavage; STZ i.c.v.	Nanoencapsulated curcumin improved memory; both forms ↓ AChE activity; NC reduced oxidative stress	Reduced neuroinflammation (GFAP); antioxidant effects; cognitive improvement

4-HNE—4-hydroxynonenal, Aβ—amyloid-beta, AChE—acetylcholinesterase, AMPKα—AMP-activated protein kinase alpha, APP—amyloid precursor protein, APPswe—Swedish mutant of APP, BDNF—brain-derived neurotrophic factor, BBB—blood–brain barrier, COX-2—cyclooxygenase-2, CREB—cAMP response element-binding protein, DCX—doublecortin, DRP1—dynamin-related protein 1, FIS1—mitochondrial fission 1 protein, GFAP—glial fibrillary acidic protein, GSH—glutathione, GSSG—glutathione disulfide; GPx—glutathione peroxidase, GR—glutathione reductase, GSK-3β—glycogen synthase kinase-3 beta, HSP—heat shock protein, IL-1β—interleukin-1 beta, iNOS—inducible nitric oxide synthase, JNK—c-Jun N-terminal kinase, LPS—lipopolysaccharide, MDA—malondialdehyde, MFN1/2—mitofusin 1/2, mTOR—mechanistic target of rapamycin, NC—nanoencapsulated curcumin, Nrf2—nuclear factor erythroid 2-related factor 2, NGF—nerve growth factor, NF-κB—nuclear factor kappa-light-chain-enhancer of activated B cells, OPA1—optic atrophy 1, PC—protein carbonyl, PGC1α—peroxisome proliferator-activated receptor gamma coactivator 1-alpha, PS1—presenilin-1, pCREB—phosphorylated CREB, ROS—reactive oxygen species, SLN—solid lipid nanoparticles, STZ—streptozotocin, TC—Tetrahydrocurcumin, TBARS—thiobarbituric acid reactive substances, Tg—transgenic, TFAM—mitochondrial transcription factor A, WT—wild type. Arrows indicate the direction of change: ↑ increase, ↓ decrease.

**Table 2 nutrients-17-02884-t002:** Preclinical studies on curcumin in experimental Parkinson’s disease models.

Reference	Model	Dose	Duration	Route	Key Effects	Proposed Mechanisms
Abbaoui et al. [222]	Wistar rats (copper-induced Parkinson’s disease)	Cu: 10 mg/kg i.p.; Curcumin: 30 mg/kg oral	3 days	i.p., oral	Restored tyrosine hydroxylase (TH) expression and locomotor function	Antioxidant and anti-inflammatory activity; protection of TH+ neurons from copper-induced neurotoxicity
Rajeswari et al. [223]	Mice (MPTP model)	MPTP: 40 mg/kg; Curcumin: 80 mg/kg; ThC: 60 mg/kg	7 days	i.p.	Restored dopamine (DA) and DOPAC levels; reduced MAO-B activity	Dopaminergic neuron preservation via MAO-B inhibition and dopamine metabolism protection
Pan et al. [224]	C57BL/6 mice (MPTP model)	50 mg/kg	5 days	i.p.	Reduced dopaminergic neuron loss; improved TH levels	Inhibition of JNK-mediated mitochondrial apoptosis
Jiang et al. [225]	SH-SY5Y cells (A53T α-syn overexpression)	6 μM	24–48 h	In vitro	Reduced α-synuclein accumulation; restored autophagic flux; improved neuronal viability	Downregulation of mTOR/p70S6K signaling; increased LC3-II; enhanced autophagosome formation
Ramires Júnior et al. [227]	Mice (rotenone model)	25–50 mg/kg	30 days	Oral	Improved motor function; preserved mitochondrial complex I activity	Oxidative stress reduction; mitochondrial protection; enhanced efficacy via nanoemulsion formulation
Yang et al. [228]	Rats (6-OHDA model)	5–20 mg/kg	37 days	Oral	Improved behavior, cognition, DA and NE levels	Activation of BDNF/TrkB/PI3K pathway; neurogenesis; monoaminergic system support
Fikry et al. [229]	Rats (rotenone model)	30 mg/kg	60 days	i.p.	Preserved cerebellar structure and motor function; reduced oxidative stress	Antioxidant defense restoration (↑GSH, ↑SOD, ↓MDA); reduced gliosis; improved acetylcholinesterase (AChE) activity
Wang et al. [226]	SH-SY5Y cells (α-syn oligomers)	4 μM	48–72 h	In vitro	Reduced ROS and apoptosis; stabilized α-syn aggregates	Caspase-3 inhibition; ROS suppression; attenuation of α-syn toxicity
Jagatha et al. [230]	N27 cells (in vitro) and GSH-deficient mice (BSO-induced)	10 μM (in vitro); 50 mg/kg i.p. (in vivo)	1–3 days	In vitro, i.p.	Restored GSH; protected mitochondrial complex I; reduced protein oxidation	γ-GCL activation; mitochondrial stabilization; redox balance normalization
Sharma et al. [231]	Rats (LPS-induced PD)	40 mg/kg	21 days	i.p.	Reduced astrocytosis, cytokines, iron deposition, and α-syn aggregation	NF-κB inhibition; antioxidant effects; apoptosis regulation; iron metabolism normalization
Liu et al. [233]	Wistar rats (6-OHDA model)	40, 80, or 160 mg/kg/day	14 days	Oral (gavage)	High dose improved motor performance; preserved TH+ neurons in substantia nigra	Dose- and time-dependent neuroprotection; dopaminergic preservation; reduced oxidative stress and inflammation
Geng et al. [234]	Male C57BL/6J mice (MPTP model)	MPTP: 30 mg/kg/day i.p.; Curcumin or Curoil: 120 mg/kg/day oral	7 days	i.p., oral	Curoil improved motor function; protected TH+ neurons in substantia nigra	Enhanced curcumin bioavailability; antioxidant and neuroprotective effects; TH+ neuron preservation

6-OHDA—6-hydroxydopamine, AChE—acetylcholinesterase, α-syn—alpha-synuclein, BDNF—brain-derived neurotrophic factor, BDNF/TrkB/PI3K: brain-derived neurotrophic factor/tropomyosin receptor kinase B/phosphoinositide 3-kinase; BSO—buthionine sulfoximine, Curcumin—diferuloylmethane, DA—dopamine, DOPAC—3,4-dihydroxyphenylacetic acid, GSH—glutathione, γ-GCL—gamma-glutamylcysteine ligase, i.p.—intraperitoneal, JNK—c-Jun *N*-terminal kinase, LC3-II—microtubule-associated proteins 1A/1B light chain 3B (lipidated form), LPS—lipopolysaccharide, MPTP—1-methyl-4-phenyl-1,2,3,6-tetrahydropyridine, MAO-B—monoamine oxidase B, mTOR—mechanistic target of rapamycin, NE—norepinephrine, NF-κB—nuclear factor kappa-light-chain-enhancer of activated B cells, ROS—reactive oxygen species, TH—tyrosine hydroxylase. Arrows indicate the direction of change: ↑ increase, ↓ decrease.

**Table 3 nutrients-17-02884-t003:** Preclinical studies on curcumin in experimental stroke models.

Ref.	Model	Dose	Duration	Route	Key Effects	Mechanisms
Li et al. [235]	MCAO in mice; neuronal cultures	100 mg/kg i.p. (in vivo); 4 μM (in vitro)	3 days (in vivo); 24 h (in vitro)	i.p., i.v.	Improved neurological function; reduced brain edema and infarct volume	Activation of AKT/GSK-3β/Nrf2 pathway; M2 microglia polarization; reduced oxidative stress
Jia et al. [236]	Transient MCAO in rats + MTM	300 mg/kg i.p. + 250 μg/kg MTM i.c.v.	24 h	i.p., i.c.v.	Reduced infarct size and oxidative stress; improved neurological outcomes	SP1-mediated upregulation of Prdx6; antioxidant protection
Altinay et al. [237]	Global ischemia (BCCAO)	300 mg/kg oral or i.p.	21 days (oral); 72 h (i.p.)	Oral, i.p.	Increased antioxidant enzymes; reduced inflammation; improved outcome	Increased SOD, CAT, GPx; reduced IL-6, TNF-α, apoptosis
Marques et al. [238]	ICH in rats	30 mg/kg	3 doses/48 h	i.p.	Improved motor recovery; reduced hematoma volume	Enhanced antioxidant activity; improved bioavailability
Ran et al. [239]	MCAO in mice	150 mg/kg	7 days	i.p.	Reduced white matter damage; improved sensorimotor function	Inhibition of NF-κB/NLRP3; reduced pyroptosis
Xie et al. [240]	MCAO in mice; OGD/R in cells	100–400 mg/kg (in vivo); 5–35 μM (in vitro)	Single dose	i.p., in vitro	Reduced apoptosis; improved mitochondrial function	Downregulation of Bax, upregulation of Bcl-2; preserved mitochondrial membrane potential
Li et al. [241]	MCAO/reperfusion in rats	300 mg/kg	Single dose, pre-reperfusion	i.p.	Reduced infarct size, brain edema, and inflammation	Inhibition of NF-κB, MMP-9, and caspase-3; BBB protection
Liu et al. [242]	MCAO in rats	300 mg/kg	7 days	i.p.	Increased neurogenesis; reduced neurological deficits	Activation of Notch signaling
Wu et al. [243]	MCAO/reperfusion in rats	300 mg/kg	Single dose, pre-MCAO	i.p.	Reduced BBB leakage; increased tight junction protein expression	Upregulation of ZO-1 and occludin; inhibition of NF-κB, MMP-9
Liu et al. [244]	Distal MCAO in mice	150 mg/kg	10 days	i.p.	Reduced infarct size; improved sensorimotor recovery; M2 microglia shift	Anti-inflammatory activity; enhanced M2 polarization
Wu et al. [245]	Primary cortical neurons (OGD/R); MCAO in rats	2.5–25 μM (in vitro); dose not specified (in vivo)	1 h OGD + 24 h reoxygenation (in vitro); 60 min MCAO (in vivo)	In vitro; likely i.p. (in vivo)	Reduced neuronal injury and oxidative stress; improved cell viability; reduced infarct volume	Activation of Akt/Nrf2 pathway; NQO1 upregulation; effect blocked by PI3K inhibitor LY294002
Jiang et al. [246]	MCAO in rats; astrocytes; brain capillary endothelial cells (BCECs)	0.5–2.0 mg/kg	Single i.v. injection, 30 min post-reperfusion (effects at 48 h)	i.v.	Reduced infarct size, BBB permeability, brain edema, and mortality; protected BCECs	BBB preservation via ONOO^−^ inhibition; suppression of iNOS/NO pathway
Zhang et al. [247]	MCAO in rats	25 mg/kg	Single dose	i.p.	Reduced brain edema and infarct size; decreased TNF-α, IL-6, p53, Bax; increased Bcl-2 and Sirt1; improved mitochondrial function	Anti-inflammatory and anti-apoptotic activity; mitochondrial protection

AKT—protein kinase B, Bax—Bcl-2-associated X protein, Bcl-2—B-cell lymphoma 2, BBB—blood–brain barrier, BCCAO—bilateral common carotid artery occlusion, BCECs—brain capillary endothelial cells, CAT—catalase, GPx—glutathione peroxidase, ICH—intracerebral hemorrhage, IL-6—interleukin-6, i.c.v.—intracerebroventricular, iNOS—inducible nitric oxide synthase, i.p.—intraperitoneal, i.v.—intravenous, MCAO—middle cerebral artery occlusion, MMP-9—matrix metalloproteinase-9, MTM—metformin, NF-κB—nuclear factor kappa-light-chain-enhancer of activated B cells, NLRP3—NOD-like receptor pyrin domain-containing protein 3, NQO1—NAD(P)H quinone oxidoreductase 1, Nrf2—nuclear factor erythroid 2–related factor 2, OGD/R—oxygen–glucose deprivation/reoxygenation, ONOO^−^—peroxynitrite, p53—tumor suppressor protein p53, PI3K—phosphoinositide 3-kinase, Prdx6—peroxiredoxin 6, Sirt1—sirtuin 1, SOD—superoxide dismutase, SP1—specificity protein 1, TNF-α—tumor necrosis factor alpha, ZO-1—zonula occludens-1.

**Table 4 nutrients-17-02884-t004:** Summary of human clinical trials investigating the effects of curcumin in Alzheimer’s disease, Parkinson’s disease, and stroke.

Ref.	Population	Dose	Duration	Route	Key Effects	Mechanisms
Small et al. [255]	40 non-demented adults (51–84 y)	180 mg/day Theracurmin^®^ (90 mg BID)	18 months	Oral	Improved verbal and visual memory; enhanced attention; reduced amyloid and tau deposition (PET)	Anti-inflammatory; anti-amyloidogenic; Akt/Nrf2 activation
Cox et al. [256]	60 healthy adults (60–85 y)	400 mg/day Longvida^®^	4 weeks	Oral	Acute: improved attention and working memory (1 h); Chronic: improved mood, reduced fatigue and LDL	Antioxidant; anti-inflammatory; modulation of lipid metabolism
Baum et al. [257]	34 probable/possible AD patients	1 g or 4 g/day	6 months	Oral	No cognitive improvement (MMSE); safe; curcuminoids detected in plasma	Possible Aβ disaggregation; antioxidant; low bioavailability; improved capsule absorption
Ringman et al. [258]	36 mild-to-moderate AD patients	2 g or 4 g/day Curcumin C3 Complex^®^	24 weeks double-blind + 24 weeks open-label	Oral	No cognitive benefit (ADAS-Cog); well tolerated	Anti-inflammatory; antioxidant; inhibition of amyloid-β aggregation
Rainey-Smith et al. [259]	96 older adults (40–90 y)	1500 mg/day Biocurcumax™ (BCM-95^®^)	12 months	Oral	Prevention of cognitive decline (MoCA) at 6 months; no other changes	Antioxidant; anti-inflammatory
Ghodsi et al. [260]	60 idiopathic PD patients (≥30 y)	80 mg/day nanomicellar curcumin	9 months	Oral	No motor or QoL improvement; mild GI adverse effects	Anti-inflammatory; anti-apoptotic
Maghbooli et al. [261]	50 idiopathic PD patients (≥35 y)	160 mg/day nanomicellar curcumin (80 mg BID)	3 months	Oral	Improved sleep quality and QoL (PDQ-39); no effect on fatigue	Neuroprotective; antioxidant; anti-inflammatory
Donadio et al. [262]	19 PD patients + 14 PD controls	2 g/day curcumin–phospholipid (Meriva^®^)	12 months	Oral	Reduced autonomic and non-motor symptoms; slowed progression; reduced phosphorylated α-synuclein deposition	BBB penetration; anti-amyloidogenic; antioxidant; anti-inflammatory
Boshagh et al. [263]	56 ischemic stroke patients in rehabilitation	500 mg curcumin + 5 mg piperine/day	12 weeks	Oral	Reduced hs-CRP, cholesterol, TG, CIMT, BP, weight, waist circumference; increased TAC	Anti-inflammatory; antioxidant; vascular plaque reduction; BP lowering; lipid profile improvement

Aβ—amyloid-β, ADAS-Cog—Alzheimer’s Disease Assessment Scale–Cognitive Subscale, AD—Alzheimer’s disease, Akt—protein kinase B, α-synuclein—alpha-synuclein, BBB—blood–brain barrier, BID—twice daily, BP—blood pressure, BCM-95^®^—Biocurcumax™ curcumin formulation, CIMT—carotid intima–media thickness, GI—gastrointestinal, hs-CRP—high-sensitivity C-reactive protein, LDL—low-density lipoprotein, Longvida^®^—lipidated curcumin formulation, Meriva^®^—curcumin–phospholipid complex, MMSE—Mini-Mental State Examination, MoCA—Montreal Cognitive Assessment, Nrf2—nuclear factor erythroid 2–related factor 2, PD—Parkinson’s disease, PDQ-39—Parkinson’s Disease Questionnaire-39, PET—positron emission tomography, QoL—quality of life, TAC—total antioxidant capacity, TG—triglycerides, Theracurmin^®^—highly bioavailable curcumin formulation, y—years.

## Data Availability

Data sharing is not applicable to this article as no new data were created or analyzed in this study.

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
