# Peer review of "The Neuroprotective Role of Curcumin: From Molecular Pathways to Clinical Translation—A Narrative Review"

_nutrients, 2025, doi:10.3390/nu17172884_

Round 1
Reviewer 1 Report
Comments and Suggestions for Authors
This manuscript by Lehoczki et al. provides a comprehensive and well-structured narrative review on the neuroprotective potential of curcumin. The review is commendable for its breadth, covering the topic from fundamental chemistry and molecular mechanisms to a detailed summary of preclinical and clinical evidence for Alzheimer's disease, Parkinson's disease, and stroke. While the manuscript is thorough, several points could be addressed to further enhance its scientific impact and completeness. My comments are detailed below.
Major Comments
- While the review effectively summarizes outcomes from various human clinical trials, Section 6 and the "Human Clinical Trials" section primarily function as a catalog of individual study results. Although the manuscript appropriately acknowledges that outcomes are "variable," it lacks the critical analysis necessary to explain these inconsistencies. The authors should develop a comprehensive synthesis examining factors that distinguish successful trials from inconclusive ones :
a. A comparative analysis should examine whether trials yielding null results predominantly utilized low-bioavailability formulations versus those demonstrating positive effects (such as Theracurmin® or Longvida®). This comparison would directly support the paper's central argument regarding bioavailability.
b. The analysis should evaluate whether trials targeting early-stage, cognitively intact populations (like the Small et al. study) achieved greater success than those involving patients with established moderate AD (such as the Ringman et al. study). Such findings could indicate curcumin's primary utility lies in prevention rather than therapeutic intervention.
c. The discussion should address whether studies demonstrating benefits employed more sensitive or distinct cognitive and biological markers compared to those showing no effect. - The mechanistic discussion omits a crucial pathway: the gut-brain axis. Given curcumin's minimal oral bioavailability, substantial quantities remain in the gastrointestinal tract, where they can significantly influence gut microbiota. The preclinical AD table references Sun et al.'s work linking curcumin's effects to gut microbiota modulation, yet this mechanism receives insufficient attention. Please address how curcumin:
- Alters gut microbiome composition
- Influences microbial metabolite production (particularly short-chain fatty acids with systemic anti-inflammatory and neuroprotective properties)
- Affects intestinal barrier integrity ("leaky gut"), which is increasingly linked to neuroinflammation. This would provide a more complete picture of curcumin's action, directly linking its poor absorption to a plausible, alternative mechanism of action.
3. Section 4.4 ("Curcumin and the Inhibition of Amyloid Aggregation") focuses appropriately on Amyloid-beta (Aβ) in AD context. However, given the review's extensive coverage of Parkinson's Disease (characterized by α-synuclein aggregation) and Tau pathology mentions, this section requires expansion.The section should be retitled "Modulation of Pathological Protein Aggregation" and include:
α-synuclein subsection: Evidence for curcumin's inhibition of α-synuclein fibrillation and stabilization of non-toxic oligomers as a key PD mechanism (currently relegated to later preclinical sections).
Tau subsection: Discussion of curcumin's potential to interfere with Tau hyperphosphorylation and aggregation, particularly given the Longvida® study showing Tau dimer reduction.
Minor Comments
Section 3 would benefit from a schematic diagram illustrating curcumin's pharmacokinetic challenges: poor absorption, rapid first-pass metabolism (glucuronidation/sulfation), and systemic elimination. This visual aid would enhance comprehension for non-pharmacology specialists.
Comments on the Quality of English Language
Final proofreading is required to address minor grammatical errors and ensure terminological consistency throughout the manuscript.
Author Response
Dear Reviewer,
We would like to sincerely thank the Reviewer for the thorough, detailed, and constructive feedback, which has greatly contributed to the further improvement of our manuscript. Below we provide our point-by-point responses to the raised issues:
- Synthesis and critical analysis of clinical trials
We have added a new section that synthesizes and critically analyzes clinical trial data. This has been included as Section 7 and is highlighted in blue in the revised manuscript. - Inclusion of the gut–brain axis among the mechanisms
In line with the Reviewer’s suggestion, we have expanded the manuscript with the following aspects:- Effects of curcumin on gut microbiota composition.
- Influence on microbial metabolites (e.g., short-chain fatty acids).
- Protection of intestinal barrier integrity (“leaky gut”) and its relevance to neuroinflammation.
- Expansion on pathological protein aggregation
We have revised Section 4.4, now entitled “Modulation of Pathological Protein Aggregation”, in accordance with the Reviewer’s suggestion. The modifications are highlighted in blue in the revised version. - Illustration of pharmacokinetic barriers
To further clarify Section 3, we have prepared and included a new figure that illustrates the pharmacokinetic challenges of curcumin (e.g., absorption, rapid elimination).
In summary, we are very grateful for the Reviewer’s constructive comments, which have significantly enhanced both the scientific value and the readability of our manuscript.
Respectfully,
Prof. János T Varga and co-authors
Reviewer 2 Report
Comments and Suggestions for Authors
The manuscript is well-written and addresses an important topic; however, several aspects could be improved to strengthen the work.
First, it is necessary to specify in the title that this is a review, indicating the type (narrative, systematic, or scoping). Likewise, the abstract does not explicitly mention that this is a review; I recommend including this information, along with the number of articles analyzed and the main objective of the review.
Although the introduction states that this is a narrative review, the guiding research question is not clearly formulated, and I suggest clarifying it.
Additionally, the rationale for restricting the temporal scope to the years 2000–2025 should be better justified.
While the manuscript is well written, the topic appears to have already been widely discussed in the literature, particularly given the broad focus on curcumin. Narrowing the scope or emphasizing a specific gap in the literature could enhance the novelty of the contribution.
Two topics have a similar heading (Limitations). Please revise it.
What is the primary novelty of this manuscript? Numerous studies have explored the use of curcumin in managing neurodegenerative diseases, including structured reviews that employ systematic methodologies and thorough data evaluations. Thus, a significant drawback is the lack of originality. While I appreciate that this study is well-written, it needs to provide a meaningful contribution to the field.
The investigation of nano-based systems incorporating curcumin presents an underexplored area in literature, particularly when compared to the more conventional discussions surrounding the application of this phytochemical in the treatment of neurodegenerative diseases. This promising avenue warrants further examination to enhance our understanding of curcumin's therapeutic potential in a nanoscale context.
Numerous studies have already investigated the safety profile of curcumin. This topic could also be further explored.
Finally, I suggest including figures to improve clarity and engagement with the reader, for example, in section 4.2 (“Anti-Inflammatory Mechanisms of Curcumin: Inhibition of NF-κB, COX-2, and iNOS”).
Current and well-structured reference framework.
Author Response
Dear Reviewer,
We sincerely thank you for your valuable and detailed feedback on our manuscript. Your suggestions have greatly helped improve the quality of the paper, and we have carefully addressed each point:
- The title and abstract now clearly indicate that this is a narrative review.
- The guiding research question has been clarified in the Introduction to make the focus and purpose transparent to readers.
- The rationale for the 2000–2025 timeframe is now explicitly explained in the Methods section.
- The scope of the manuscript has been refined, emphasizing the gaps in the literature that our study addresses.
- The two “Limitations” headings have been revised to clearly distinguish them.
- We have highlighted the novelty and contribution of our study, particularly regarding the investigation of curcumin in nano-based systems, which is underexplored in current literature.
- The section on the safety profile of curcumin has been expanded with relevant findings.
- Following your suggestion, two new figures have been added (e.g., in section 4.2 illustrating the anti-inflammatory mechanisms) to enhance clarity and readability.
- The reference framework has been reviewed and structured to ensure it remains current and relevant.
We hope that these revisions meet your expectations and that the updated manuscript clearly demonstrates the novelty and scientific value of this narrative review.
Thank you again for your constructive and supportive comments.
Sincerely,
Prof. János T Varga and co-authors
Round 2
Reviewer 2 Report
Comments and Suggestions for Authors
My comments were adequately addressed. I have no further suggestions.